# Room temperature catalytic upgrading of unpurified lignin depolymerization oil into bisphenols and butene-2

Elena Subbotina[1,2,3] ✉, Layra Rodrigues Souza[3], Julie Zimmerman [3,4,5] &
Paul Anastas[1,3,4,5,6] ✉

Lignin is the largest source of renewable aromatics on earth. Despite numerous techniques for lignin depolymerization into mixtures of valuable monomers, methods for their upgrading into final products are scarce. The state of the art upgrading methods generally rely on catalytic funneling, requiring high temperatures, catalyst loadings and hydrogen pressure, and lead to the loss of functionality and bio-based carbon content. Here an alternative approach is presented, whereby the target monomers are selectively converted in unpurified mixtures into easily separable final products under mild conditions. We use reductive catalytic fractionation of wood to convert lignin into iso-eugenol and propenyl syringol enriched oil followed by an olefin metathesis to yield bisphenols and butene-2, thus, valorizing all bio-based carbons. To further demonstrate the synthetic utility of the obtained bisphenols we converted them into polyesters with a high glass transition temperature ($T_g = 140.3\,°C$) and thermal stability ($Td_{50\%} = 330\,°C$).

Upgrading of biomass into commodity chemicals, polymer precursors and fuels is gaining considerable attention due to the urgent need for a sustainable petroleum-free economy. Wood is the most available source of biomass on earth, and lignin, constituting up to 30 wt% of wood, is the largest source of aromatics present in nature. Significant advances in lignin valorization strategies[1–5] offer numerous techniques to generate mixtures of value-added molecules via oxidative[6–9], reductive[2,10,11], redox-neutral[12–16], and acid-catalyzed[17–19] pathways. However, protocols for their direct upgrading into final products remain elusive. The upgrading is impeded by the multicomponent nature of lignin-derived streams, which generally necessitates cumbersome separations of individual components, and can cause catalyst poisoning by the impurities.

A possible solution to this problem is so-called chemical funneling, converting a complex mixture into a single product[20–22]. A successful demonstration of this approach was reported by the Sels

research group[20] by converting a lignin depolymerization oil mainly comprising propenyl syringol and propenyl guaiacol, obtained via reductive catalytic fractionation of wood (RCF), into phenol and propene – products with a currently existing market. The transformation proceeded via a demethoxylation ($Ni/SiO_2$, 285 °C) followed by a dealkylation step (410 °C, zeolite) (Fig. 1a). Similarly, the Barta group[21] reported funneling a mixture of lignin-derived monomers (mainly propanol guaiacol and propanol syringol) into 4-(3-hydroxypropyl) cyclohexan-1-ol via hydrotreatment (Raney Ni, 150 °C, 30 bar $H_2$). The product was purified via column chromatography and used to prepare bio-based polyethylene terephthalate (PET) analogs (Fig. 1b). While a powerful technique, chemical funneling inherently implies de-functionalization, leading to the diminished bio-based carbon content or loss of aromaticity in the final product, and typically features harsh reaction conditions as well as a need for an external hydrogen source. Relatedly, Epps III et al. reported a direct conversion of a mixture of

[1]Department of Chemistry, Yale University, 225 Prospect St, New Haven, CT, USA. [2]Department of Fibre and Polymer Technology, Wallenberg Wood Science Center, KTH Royal Institute of Technology, Teknikringen 56, 100 44 Stockholm, Sweden. [3]Center for Green Chemistry & Green Engineering at Yale, 370 Prospect St, New Haven, CT, USA. [4]Department of Chemical and Environmental Engineering, Yale University, 17 Hillhouse Ave, New Haven, CT, USA. [5]Yale School of the Environment, 195 Prospect St, New Haven, CT, USA. [6]Yale School of Public Health, 60 College St, New Haven, CT, USA.
✉e-mail: elenasu@kth.se; Paul.anastas@yale.edu

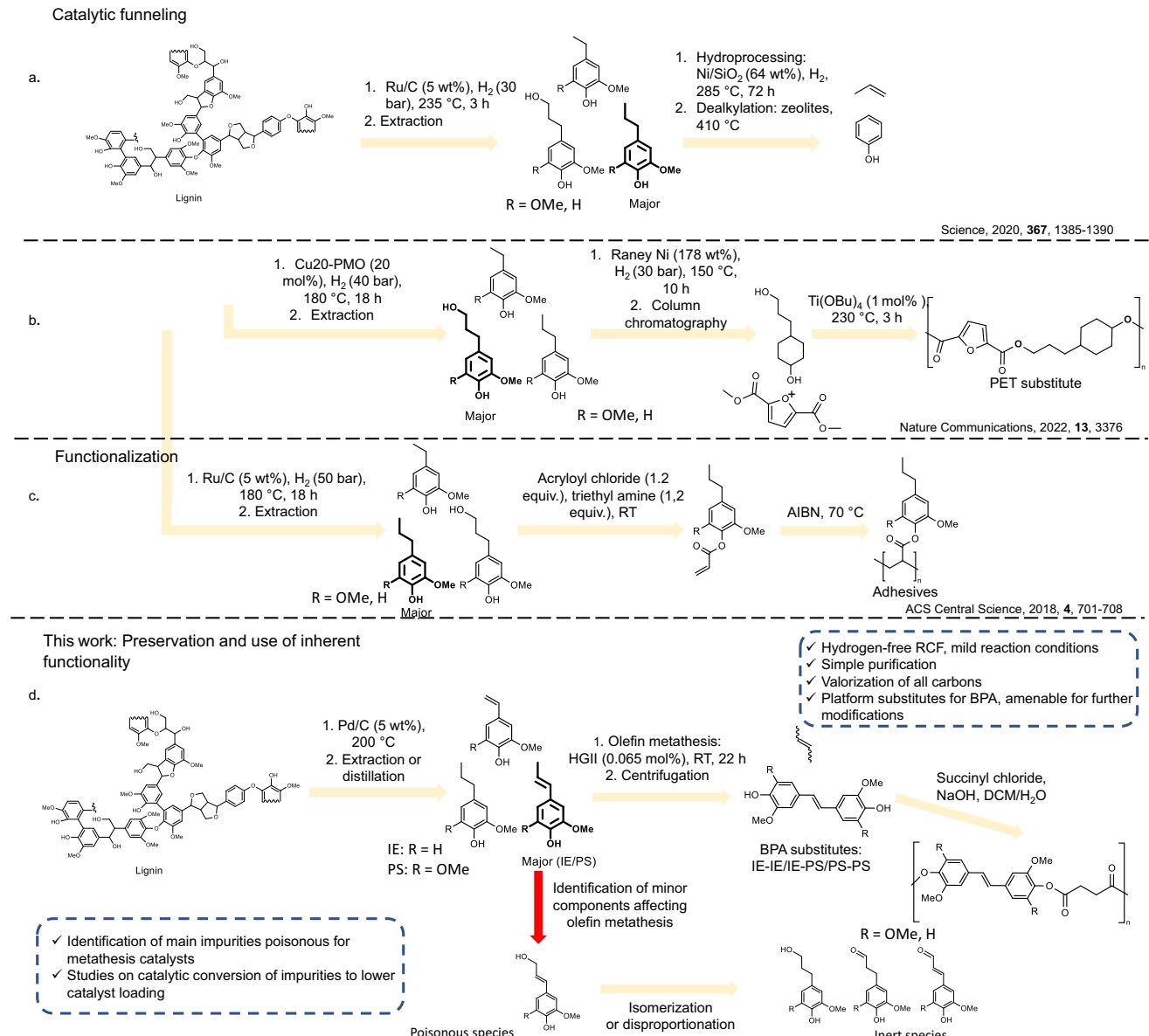

**Fig. 1 | Strategies for upgrading of lignin-derived monomers oil. a** Previous work: Catalytic funneling of lignin-derived monomers oil, enriched in 4-propylguaiacol and 4-propylsyringol into phenol and propylene via a two-step process (hydroprocessing/dealkylation). **b** Previous work: Catalytic funneling of lignin-derived monomers oil enriched in 4-propanolguaiacol and 4-propanolsyringol into 4-(3-hydroxypropyl) cyclohexan-1-ol via Raney nickel catalyzed reduction and its further conversion into bio-based polyethylene terephthalate (PET) analog. PMO porous metal oxide. **c** Previous work: Functionalization of lignin-derived monomers: acryloylation of lignin-derived mixture of monomers (mainly 4-propylguaiacol and 4-propylsyringol) and its polymerization into pressure-sensitive adhesives. AIBN Azobisisobutyronitrile

**d** This work: Preservation and use of inherent functionality of lignin monomers. The developed process includes: reductive catalytic fractionation of wood (RCF); hexane extraction of lignin-derived monomers oil, enriched in iso-eugenol (IE) and propenyl syringol (PS); catalytic conversion of lignin-derived monomers oil (mainly IE and PS) into bisphenols IE-IE, IE-PS, PS-PS (substitutes of bisphenol A−BPA) and butene-2 via olefin metathesis; isolation (simple centrifugation) of the bisphenols and their polyesterification with succinyl chloride into final materials. In addition, studies of the effect of impurities present in lignin monomers oil on olefin metathesis step were performed, and protocols for their transformation into catalytically inert species were developed. HGII Hoveyda Grubbs 2nd generation catalyst.

4-propyl syringol and 4-propyl guaiacol obtained from wood via RCF into the corresponding acrylates by an esterification with acryloyl chlorides followed by a reversible addition−fragmentation chain-transfer (RAFT) polymerization to yield pressure sensitive adhesives (Fig. 1c)[23]. In contrast to the prior art, in this approach a new functionality was added to the products (rather than removed) and further used for the conversion of the modified monomers into the final materials. While these examples signify a considerable advancement in the field, in an idealistic scenario for upgrading of depolymerized lignin oil, all bio-based carbons are preserved in the final products,

addition of non-biobased carbons is avoided, purification steps are minimized, and reaction conditions are mild.

Herein a catalytic conversion of unpurified lignin depolymerization oil into bisphenols and butene-2 at room temperature without stoichiometric reagents and tedious separations is demonstrated. This work is building upon the greener hydrogen-free version of RCF, developed by Samec and coworkers, delivering oils enriched in iso-eugenol (IE) and propenyl syringol (PS)[24]. It is envisioned to convert this mixture into the corresponding bisphenols (PS-PS, IE-IE and PS-IE, Fig. 1d) via metathesis reaction with a concomitant generation of

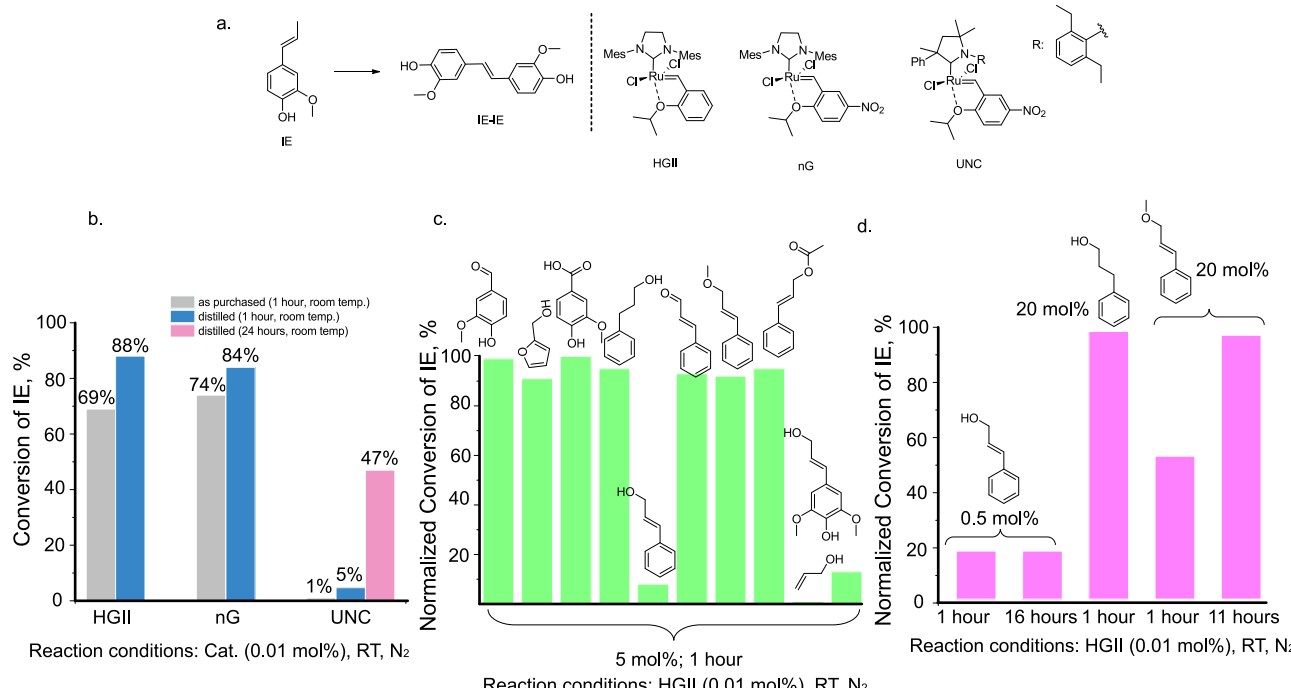

**Fig. 2 | Self-metathesis of iso-eugenol (IE) in its pure form and in the presence of various additives. a** A reaction scheme of self-metathesis of IE and structures of the screened catalysts: Hoveyda Grubbs 2nd generation catalyst (HGII), nitroGrela (nG), UltraNitroCat (UCN). **b** Catalytic performance of the catalysts. Reaction conditions: 0.01 mol% catalyst, room temperature, $N_2$ atmosphere, presented yields (conversions) are average of at least two repetitions. **c** Effect of various additives on a catalytic performance of HGII in self-metathesis of IE. Reaction conditions: 0.01 mol% catalyst, 5 mol% additive, room temperature, $N_2$ atmosphere, 1 h. Normalized conversion = (conversion in the presence of an additive)/(conversion in the absence of an additive)*100% (average of at least two repetitions). Sinapyl alcohol was added in 4.3 mol%. **d** Effect of various amounts of additives on a catalytic performance of HGII in self-metathesis of IE. Reaction conditions: 0.01 mol% catalyst, room temperature, $N_2$ atmosphere. Normalized conversion based on at least two repetitions.

butene-2[25]. To enable the catalytic step directly on lignin oil, a thorough investigation of the effects of other components of the oil on the catalytic performance is reported and means for their in situ conversion into inert species is demonstrated. This approach allows the valorization of all bio-based carbons into industrially relevant products; for example, the bisphenols are candidates for bisphenol-A (BPA) substitutes[26,27]. BPA is widely used as a monomer for the production of polyesters and polycarbonates and as a plasticizer[28]. However, the toxicological profile and fossil oil origin of BPA raise significant concerns regarding its commercial uses and increase demands for substitutes[29–31]. Note, that the obtained here bisphenols are highly functionalized and can be considered as platform molecules, amenable for further diversification. In addition, butene-2 is a valuable light olefin, currently generated from crude oil for polymer and pharmaceutical industries[32]. Production of butene from biomass is largely undeveloped, and generally relies on energy-intensive processes, such as Fischer–Tropsch to olefins (FTO) or methanol to olefins (MTO)[33]. Overall, an entire process chain for the conversion of wood into functional polymeric materials and light olefins is reported here. More generally, this work opens avenues for application of transition metal catalysis directly to complex unpurified bio-derived streams via a rational system design.

## Results

### Self-metathesis of IE and effect of the impurities

To enable olefin metathesis on unpurified lignin oil we decided to test ruthenium-based olefin metathesis catalysts with increased stability and rate of initiation, such as phosphine-free Hoveyda Grubbs II generation (HGII) and nitroGrela (nG). In addition, we evaluated a commercial cyclic alkyl amino carbene (CAAC)-based catalyst UltraNitroCat (UCN) for its known increased air and moisture tolerance (Fig. 2a–b). The reactions were performed with IE as a substrate.

IE is a convenient substrate for initial screening and is of interest not only for lignin valorization but also for other types of biomasses as it is a major component of essential oils (e.g. clove oil). We have not performed a comprehensive optimization of the reaction conditions on IE, which would not be directly transferable to a realistic lignin-derived mixture, but rather limited ourselves to a short catalyst screening.

The reaction was performed neat, at room temperature ($N_2$ atmosphere, glove box) using 0.01 mol% of the catalyst with IE as purchased. Both HGII and nG showed comparable performance (although nG expectedly exhibited faster initiation), and the product was formed in 68% and 74% yield for HGII and nG respectively after 1 h (Fig. 2a). A faster initiation in case of nG is caused by the electron withdrawing effect of the nitro group, which facilitates the dissociation of isopropoxy group from the metal center, which is directly involved in the initiation step. When a freshly distilled IE was used the yield improved and reached 84-88%, which can be rationalized by the presence of oxygenated species in as purchased IE (Sigma-Aldrich, 98%, mixture of cis and trans isomers) acting as catalysts' poisonings (Fig. 2a). No by-products were detected in the reaction mixture and the conversion of IE was equal to the yield of IE-IE. Unlike HGII and nG, under standard conditions UNC only allowed for the formation of trace amounts of the product. Using a freshly distilled substrate or significantly longer reaction time (36 h) did not lead to the formation of product in a desirable yield (the product was formed in 47% yield). While we have not performed any further experimental studies regarding the lower reactivity of UNC, based on previous literature reports, we believe steric factors might play a role in the observed catalytic behavior. UNC is more sterically congested compared to nG and HGII, which can slow down the initiation. For the initiation of the tested catalysts isopropoxy group of benzylidene moiety needs to dissociate from ruthenium, and the benzylidene moiety needs to acquire a position in a plane parallel to Ar-N bond in order to provide a

space for the incoming olefin, which is sterically unfavorable in case of UNC[34,35]. The steric considerations might become even more pronounced for PS due to an additional methoxy group. Based on these results, HGII was chosen for further studies.

To enable this reaction on unpurified lignin oil, the effects of other common lignin-derived monomers generally accompanying target IE and PS, featuring hydroxyl, carboxyl, aldehyde or olefinic functionalities were studied (Fig. 2c). Addition of 5 mol% of vanillin, vanillic acid or cinnamyl aldehyde to IE did not lead to any significant drop in the yield of the target homodimer (Fig. 2c). As primary alcohols are known to cause degradation of Grubbs 1st generation (GI)[36,37] and Grubbs 2nd generation (GII) catalyst[38], the effects of 3-phenyl propanol and furfuryl alcohol were tested. Neither of the alcohols caused a significant change in product yield (Fig. 2c). Even higher loadings of 3-phenyl propanol (20 mol%) did not cause significant drop in yield of IE-IE (Fig. 2d). In addition, the reaction mixture was analyzed by GC-MS and we did not observe any by-products derived from 3-phenyl propanol (Supplementary Fig. 1).

While saturated primary alcohols did not cause a significant drop in yield of the desired product, allylic alcohols had a clear effect. Cinnamyl alcohol, allyl alcohol and sinapyl alcohols significantly decreased the yield of the product, when they were added in 4-5 mol% to the substrate (Fig. 2c). Allylic alcohols, thus, represent a main concern for the reaction applied to lignin-derived oils, as they were found to be primary monomeric species released into the solution during RCF of wood and with a high probability can be present in the final mixture of monomers[39,40]. Even as little as 0.5 mol% of cinnamyl alcohol was found to decrease the yield of the desired product by 82% (Fig. 2d). While there are examples in literature where metathesis of allyl alcohols was performed by GII[41], as well as by HGII[42], the reactions were usually carried out using high catalyst loadings (up to 5 wt%).

To get an increased understanding of the effect of allylic alcohols on the transformation, their derivatives were tested. Cinnamyl methyl ether, cinnamyl acetate and cinnamyl aldehyde (5 mol%) had little effect on the yield of IE-IE (Fig. 2c). Larger quantities of cinnamyl methyl ether (20 mol%) slowed down the reaction resulting in 54% lower conversion after 1 h (compared to the reaction in the absence of an additive). However, the catalyst remained active, and the conversion reached the same level as in case of pure IE after 11 h (Fig. 2d). GC-MS analysis revealed the formation of products derived from self- and cross-metathesis of methyl cinnamyl ether and no products of the decomposition of cinnamyl methyl ether were observed (Supplementary Fig. 2). Thus, the lower reaction rate in this case can be attributed to lower total catalyst loading (since the total amount of olefinic species increased by 20 mol%) and different kinetics of the cross- and self-metathesis of methyl cinnamyl ether.

## Mechanistic investigation of HGII decomposition by allylic species

To gain a further insight into the influence of allylic alcohols and their derivatives the decomposition of HGII in their presence (4 mol% relative to the substrate in toluene-d8) was monitored by [1]H NMR. The decomposition was tracked by the disappearance of the alkylidene signal at 16.51 ppm (Fig. 3a). In case of cinnamyl alcohol, the alkylidene signal was almost completely lost after 80 min (Fig. 3a). With cinnamyl methyl ether, the signal was still present after 260 min. However, this signal almost fully disappeared after 22 h, instead two new signals at ~13.3 ppm (minor) and at ~13.8 ppm (major) appeared. The signal at ~13.8 ppm most certainly corresponds to Fisher carbene species ([Ru] = CHOMe) (vide infra)[43,44]. In case of cinnamyl aldehyde alkylidene signal was still present after 48 h. Importantly, no signal in the region from -26 ppm to 0 ppm for any of the samples was observed, indicating that ruthenium hydride species were either not formed in detectable by [1]H NMR amounts or were rapidly consumed.

In a bid to get additional information regarding the decomposition pathway, the reaction mixtures were analyzed by GC-MS. In the reaction mixture containing cinnamyl alcohol a substantial amount of stilbene and cinnamyl aldehyde in addition to unreacted cinnamyl alcohol was observed (Supplementary Fig.3). In case of cinnamyl methyl ether, besides the peak corresponding to the starting material and stilbene two peaks with the same molecular weight as starting material (Mw = 148), but different retention times were observed (Supplementary Fig. 4). It is assumed that these signals correspond to cis and trans enol ethers formed from methyl cinnamyl ether via an isomerization. [1]H NMR spectra of the reaction mixtures after 48 h supported this assumption; new signals were observed in the range characteristic to enol ethers: 4.5 ppm (cis-), 4.8 (trans-) ppm (PhCH2**CH**CHOMe), and 6.0-6.4 ppm (cis- and trans- PhCH2CH**CH**OMe) (Fig. 3a–b)[45]. This evidence is consistent with the isomerization of methyl cinnamyl ether to the corresponding cis and trans forms of 3-Methoxy-2-propenyl-benzene (MPB), which in turn can form Fisher carbenes with HGII. The reaction mixture containing cinnamyl aldehyde showed almost solely a peak of the starting material (Supplementary Fig. 5).

Subsequently the decomposition of HGII by cinnamyl methyl ether and cinnamyl alcohol at higher temperatures (4 mol%, 110 °C) was studied. In this case, the decomposition was significantly faster, and a complete disappearance of alkylidene signal was observed for both substrates within an hour (Fig. 3b). In case of cinnamyl methyl ether, while a significant amount of cis- and trans-MPB was detected in the reaction mixture by [1]H NMR, signals corresponding to Fisher carbenes (13-14 ppm) were absent, likely due to their more rapid decomposition (Fig. 3b). Analysis of the reaction mixture by GC-MS further confirmed a significant amount of cis- and trans-MPB (Supplementary Fig. 6). Important to mention, that under this conditions cinnamyl alcohol was partly isomerized to 3-phenyl propionaldehyde (Supplementary Fig. 7).

To identify the fate of the decomposed catalyst, ESI-HRMS (high resolution electrospray ionization mass spectroscopy) analysis was attempted of the following reaction mixtures: HGII + IE, HGII+cinnamyl alcohol and HGII+allyl alcohol (Supplementary Fig. 8). The reaction mixtures containing cinnamyl and allyl alcohol revealed a presence of high molecular weight species (Mw in a range of 900–1400 Da), which were not observed in case of IE. Importantly, there is a great degree of similarity in ruthenium species found in the reaction containing allyl alcohol and cinnamyl alcohol, suggesting that both alcohols probably cause the decomposition via a similar pathway. While it is difficult to discern a precise structure of the products of the decomposition of the catalyst which will require further studies, it is likely that the decomposition involves the formation of higher molecular mass ruthenium species.

The important implications of this part of the study are: (1) at room temperature allyl ethers cause the decomposition of HGII at significantly lower rate than allylic alcohols. However, under prolonged reaction times, they isomerize into enol ethers which cause the formation of catalytically inactive Fisher carbenes. (2) At high reaction temperatures (110 °C) and high catalyst loadings (4 mol%) the isomerization of allyl ethers proceeds significantly faster and causes a fast deactivation of HGII. (3) At higher temperature (110 °C) and high catalyst loading (4 mol% HGII) allylic alcohols convert into the corresponding saturated aldehydes. Taking into account previously proposed mechanism for the decomposition of GI by allylic species (Fig. 4b)[46], an increased poisonous effect of allylic alcohols vs. allylic ethers may be due to the increased propensity of allylic alcohols for β-hydride shift and/or existence of alternative decomposition pathways. E.g. the ruthenium alkylidene can be lost via dehydrogenation of cinnamyl alcohol (Supplementary Fig. 35), which is in line with the observed formation of cinnamyl aldehyde and previous reports[36]. However, more investigations will be needed to draw final conclusions.

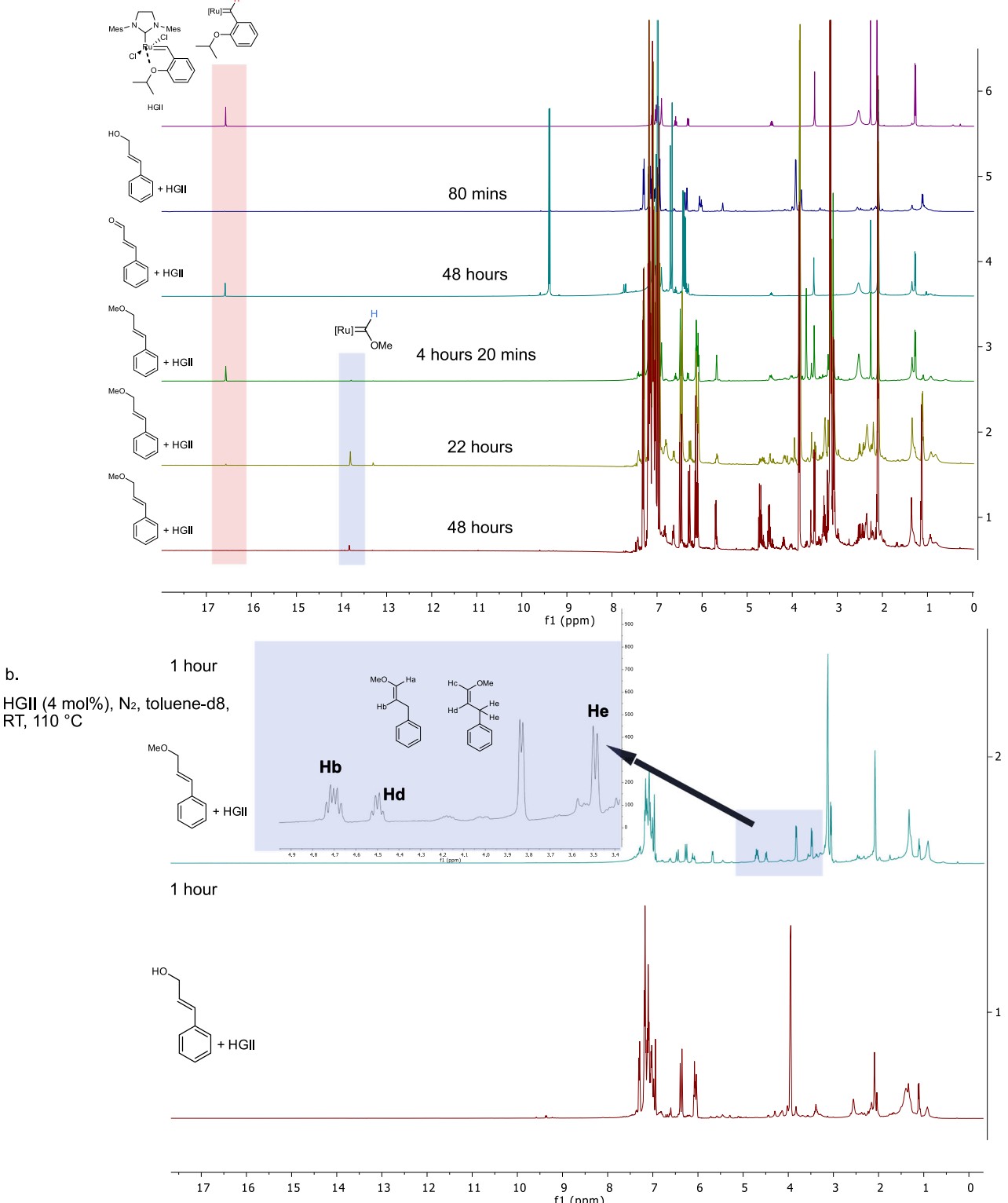

**Fig. 3 | NMR study of the deactivation of HGII (Hoveyda Grubbs 2nd generation catalyst) by cinnamyl alcohol, cinnamyl methyl ether and cinnamyl aldehyde.** **a** From top to bottom: ¹H NMR spectra of HGII in toluene-d8 and the reaction mixtures containing cinnamyl alcohol, cinnamyl aldehyde and cinnamyl methyl ether in the presence of 4 mol% HGII (N₂ atmosphere, room temperature) after specified period of time. The results indicate a rapid decomposition of HGII (disappearance of alkylidene signal at 16.51 ppm) in case of cinnamyl alcohol and

slower decomposition in case of methyl cinnamyl ether. The ¹H NMR spectra of the reaction mixture containing methyl cinnamyl ether and HGII reveal the presence of enol ethers and Fisher carbenes. **b** ¹H NMR spectra of the reaction mixtures of cinnamyl methyl ether (top) and cinnamyl alcohol (bottom) containing 4 mol% HGII, N₂ atmosphere, 110 °C, 1 h. In both cases the catalyst was decomposed (alkylidene signal was lost) within 1 h.

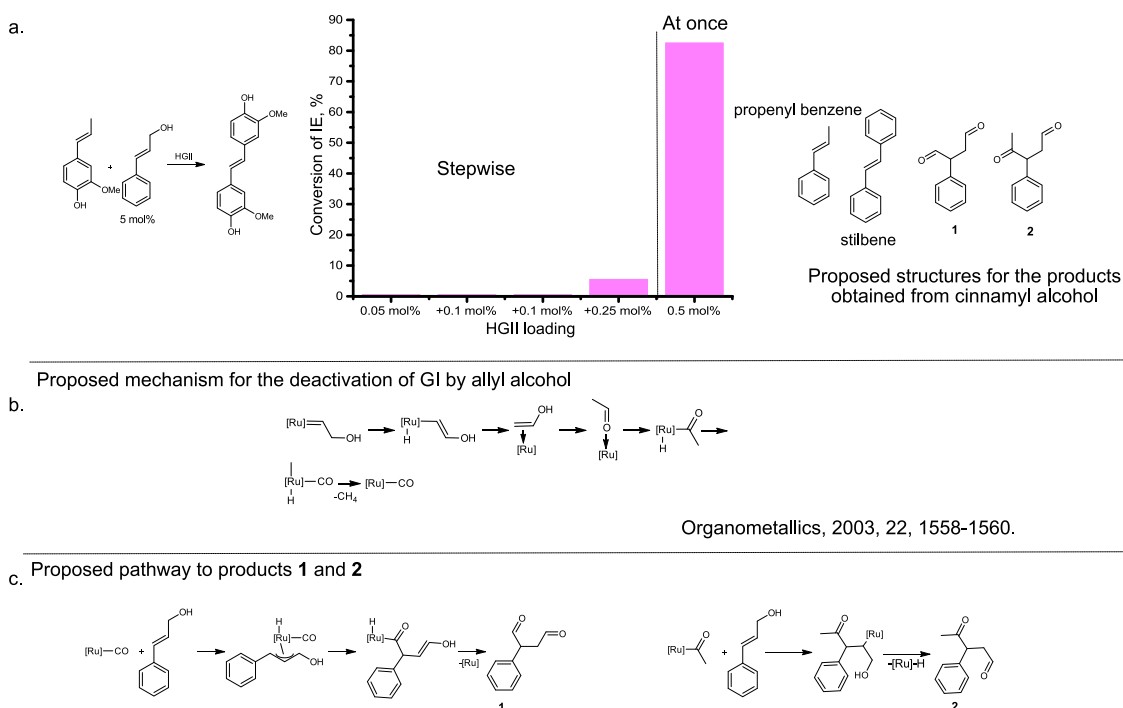

**Fig. 4 | Analysis of the reaction mixtures of self-metathesis of iso-eugenol (IE) in the presence of cinnamyl alcohol (5 mol%). a** Self-metathesis of IE in the presence of cinnamyl alcohol (5 mol%), stepwise vs. at once addition of Hoveyda Grubbs 2nd generation catalyst (HGII). Reaction conditions: room temperature, $N_2$ atmosphere, presented yields are average of at least two repetitions. During a stepwise addition of HGII to the reaction mixture of IE and cinnamyl alcohol (5 mol%) very low conversion (6%) of IE was observed even when the total catalyst loading reached 0.5 mol%. In case, when the catalyst was added in a single portion (0.5 mol%) 83% conversion of IE was achieved within 1 h. **b** A reaction mechanism proposed by Werner's group for the deactivation of Grubbs 1st generation catalyst (GI) by allylic alcohol[46]. **c** Proposed pathway for the formation of products **1** and **2** from cinnamyl alcohol during olefin metathesis of IE (5 mol% of cinnamyl alcohol, HGII 0.5 mol% added at once).

It might be difficult to avoid a minor presence of allylic alcohols in lignin-derived steams (as they are the primary species released into the solution during lignin depolymerization). Allyl alcohols can significantly increase required catalyst loadings for the metathesis. E.g., to achieve 83 mol% conversion of IE in self-metathesis (1 h) in the presence of 5 mol % cinnamyl alcohol, 0.5 mol% of HGII was required, which is 50 times more compared to pure IE (Fig. 4a). Interestingly, when the same reaction was performed with the catalyst being added in a stepwise manner, a very low conversion was achieved when the total catalyst loading reached 0.5 mol% (6% vs. 83%). GC-MS analysis revealed that in case when HGII was added in one portion almost no cinnamyl alcohol was present in the final mixture (Supplementary Fig. 9). It was converted into stilbene (a product of self-metathesis), propenyl benzene (a product of cross metathesis with IE) and products **1** and **2** (Mw = 162 and Mw = 176 respectively, for proposed structures see Fig. 4a). The formation of these products can be rationalized based on the mechanism proposed by Werner's group for the decomposition of GI by allylic species (Fig. 4b, c; for the general mechanism of olefin metathesis see Supplementary Fig. 34)[46]. On the contrary, for the mixture with a stepwise addition of HGII cinnamyl alcohol was still observed in the final mixture in large quantities (Supplementary Fig. 10). These observations are consistent with the conclusion that higher catalyst loadings not only increase the rate of IE metathesis, but also the rate of decomposition of cinnamyl alcohol, which is probably catalyzed by the decomposed HGII. These considerations prompted us to optimize in situ conversion of cinnamyl alcohol into catalytically inert species.

A one-pot tandem catalytic procedure was envisioned where cinnamyl alcohol is converted into a saturated aldehyde via an isomerization or into a saturated alcohol and α, β-unsaturated aldehyde via a disproportionation followed by self-metathesis of IE. Tandem metathesis/isomerization of allylic alcohols was realized by Snapper

and coworkers[47] who performed the metathesis reaction at room temperature (0.5 mol% catalyst) followed by an isomerization of the product at higher temperatures (200 °C). It is hypothesized that these transformations can be realized in a reverse sequence, where the isomerization of allylic alcohols is taking place at elevated temperature followed by metathesis of IE at room temperature. This sequence was tested on a mixture of IE and cinnamyl alcohol (5 mol%), using 0.025 mol% HGII (relative to IE) at 90 °C. The conversion was monitored by GC-MS (Fig. 5a). When almost full conversion of cinnamyl alcohol (primarily to 3-phenyl propionaldehyde) was achieved (12–24 h), the reaction mixture was cooled to room temperature and an additional portion of HGII (0.025 mol%) was added to carry out metathesis of IE. The final product was formed in 89% yield in 1 h with a total catalyst loading of 0.05 mol%, which is 10 times less compared to a single olefin metathesis reaction (Fig. 4a). A slightly higher catalyst loading for the metathesis of IE (0.025 mol% vs. 0.01 mol%) compared to pure IE can be rationalized by the fact that minor amounts of cinnamyl alcohol were still present in the reaction mixture.

In addition, carbonyl-chloro-hydrido-tris-(triphenylphosphine)-ruthenium(II) (RuH) was tested for the tandem isomerization/metathesis sequence. We found that in this case cinnamyl alcohol primarily undergoes a disproportionation into 3-phenyl propanol and cinnamyl aldehyde (Fig. 5b). A 92% conversion of IE was achieved in 1 hour upon addition of 0.02 mol% HGII, which corresponds to 0.047 mol% total ruthenium catalyst loading. While RuH demonstrated a good performance in the isomerization of cinnamyl alcohol in the presence of IE, it is important to mention that RuH species can slow down metathesis of IE, when present in large amounts (see Section 1 in Supplementary Information).

Thus, we have developed two alternative systems for the elimination of allylic alcohols from the reaction mixture: via either

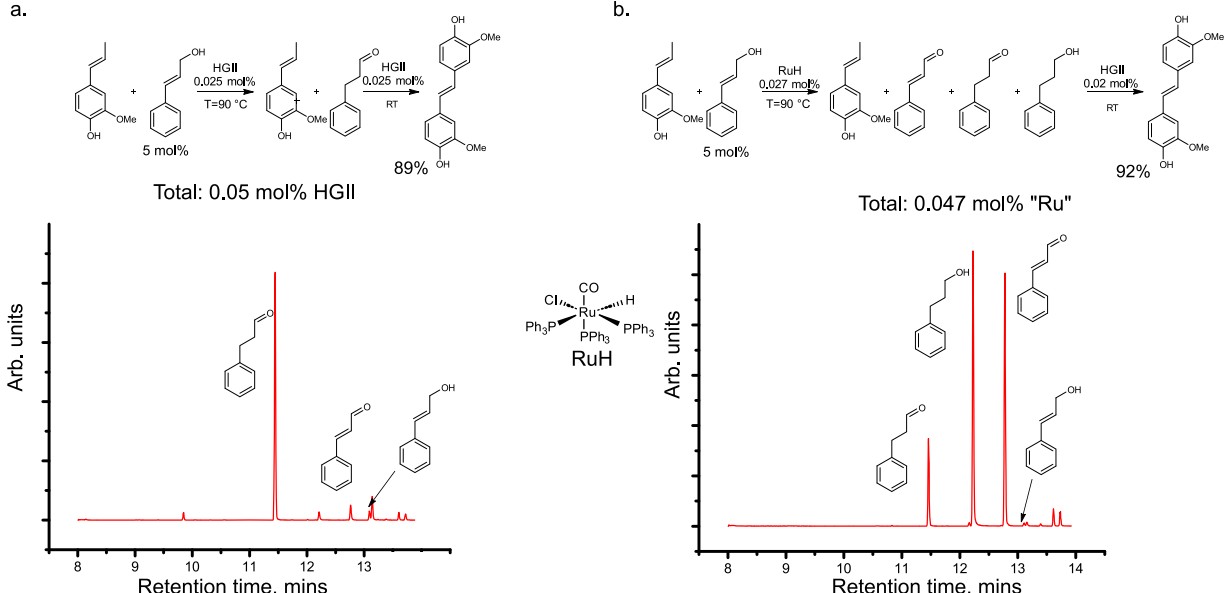

**Fig. 5 | Tandem isomerization(disproportionation)/metathesis of iso-eugenol (IE)/cinnamyl alcohol (5 mol%) mixtures. a** Conversion of cinnamyl alcohol (mainly via isomerization) (5 mol% in IE) in the presence of Hoveyda Grubbs 2$^{nd}$ generation (HGII) at 90 °C followed by the olefin metathesis of IE under standard conditions (room temperature, 0.025 mol% HGII, N$_2$). **b** Conversion of cinnamyl alcohol (mainly disproportionation) in the presence of RuClH(CO)(PPh$_3$)$_3$ (RuH) at 90 °C followed by the olefin metathesis of IE under standard conditions (room temperature, 0.02 mol% HGII, N$_2$). Both methods allowed to reach high in situ conversion of cinnamyl alcohol, which in turn allowed to perform metathesis of IE with low catalyst (HGII) loading in a second step.

disproportionation or isomerization. The development of metal-free protocols for these pathways is of interest for future studies.

The studies of the decomposition of HGII (vide supra) revealed that at higher temperature and higher catalyst loading cinnamyl alcohol ethers isomerize into the corresponding vinyl ethers, which in turn can poison the catalyst. Thus, we investigated if the designed tandem protocol is effected by the presence of cinnamyl ethers. The reaction was performed under the same reaction conditions as on Fig. 5a, but with an addition of 2.5 mol% of cinnamyl methyl ether. The isomerization of methyl cinnamyl ether was taking place, however, it was observed at a much lower extent than in case of cinnamyl alcohol (Supplementary Fig. 11). The subsequent metathesis of IE exhibited slower kinetics (the completion was reached over 20 h) and required slightly higher total catalyst loading (0.058 mol% total) to drive the reaction to 87% conversion of IE (74% was achieved with 0.05 mol%). Overall, this result demonstrated that the developed method was effective for the reaction mixtures containing allylic ethers, however, required a slightly higher catalyst loading.

### Self- and cross-metathesis of PS

The major product of the hydrogen-free RCF of hardwood is PS. Thus, after the reaction was optimized on easily available IE the reaction conditions were tested on PS. Under the reaction conditions developed for IE (0.01 mol% HGII, room temp., N$_2$ atmosphere, neat) the product was formed in 23% yield after 1 h, after 3 h the yield increased to 37%, and reached 52% after 24 h (Fig. 6a). When the reaction was performed in toluene the conversion reached 68% after 3 h and 93% after 28 h (Fig. 6a). It is apparent that self-metathesis of PS proceeds significantly slower than self-metathesis of IE. A higher conversion of PS in a solution compared to neat conditions is very likely due to a better mass transfer at the later stages of the reaction, when the reaction mixture becomes viscous and eventually solidifies. In the case of IE, due to a rapid conversion, this effect is less pronounced.

With 0.02 mol% of HGII 93% yield can be achieved already in 1 h (Fig. 6a). Interestingly, upon addition of an equimolar amount of IE the conversion of PS improved, reaching 73% in 1 h, and 92% in 10 h with 0.01 mol% catalyst loading (relative to total amount of PS and IE).

To understand the nature of the observed reactivity a series of reactions with a mixture of PS and IE in different ratios (Fig. 6b) were performed. In the equimolar PS/IE mixture (Table in Fig. 6b, entries 1–3), a slightly faster conversion of IE compared to PS was observed, but this difference was significantly diminished compared to pure substrates. Even more important observation is the evolution of the ratio between homo- and heterodimers of PS (PS-PS/PS-IE). The ratio of dimers was estimated via $^1$H NMR (Supplementary Fig. 12). At a lower conversion of starting materials there is a clear preference for cross-metathesis of PS with IE vs. self-metathesis, which is reflected in PS-IE/PS-PS ratio being > 2 times greater than the predicted ratio for random coupling (Random coupling was simulated in python, see Supplementary Information, Section 4) (Table in Fig. 6b, entry 1). Over the course of the reaction the proportion of PS-PS dimer increases getting closer to (but still below) the predicted ratio for the random coupling. The same trend was observed for the reactions performed at PS/IE ratios 0.66. 1.56 and 3.56 (Table in Fig. 6b, entries 4–7). It can be concluded that in all instances there is a preference towards the formation of a heterodimer (PS-IE) over the homodimer (PS-PS).

The enhanced reactivity of PS in the presence of IE can be rationalized via both electronic and steric factors. An increased steric demand of PS may result in a slower initiation. In addition, because of the electron-rich nature of PS the corresponding ruthenium carbene complex is more stable and thus less reactive towards ruthenacycle formation. The presence of IE leads to a faster initiation and formation of the corresponding ruthenium carbene, which in turn can react with PS to give the heterodimer.

Based on these observations, it is apparent that a presence of IE in the mixture of PS significantly facilitates the conversion of PS. As such an addition of IE to the lignin-derived oil with low IE/PS ratios is advantageous.

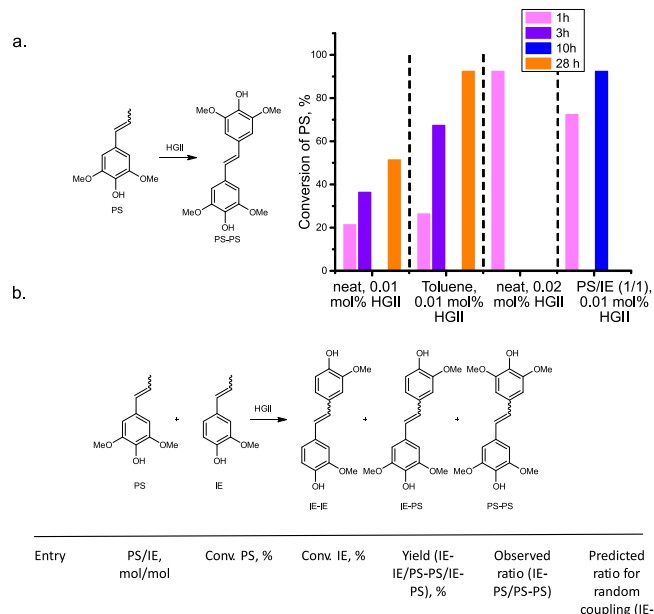

**Fig. 6 | Self- and cross-metathesis of propenyl syringol (PS). a** Self-metathesis of PS. Reaction conditions: Hoveyda Grubbs 2nd generation catalyst (HGII), N₂ atmosphere, room temperature. Addition of solvent allows to reach higher conversion due to a better mass transfer at the later stages of the reaction. Addition of equimolar amount of iso-eugenol (IE) significantly improves conversion of PS. **b** Self- and cross-metathesis of PS and IE. Reaction conditions: HGII (0.01 mol% relative to the total amount of IE and PS), N₂ atmosphere, room temperature. The results clearly indicate a higher rate of PS cross-metathesis vs. self-metathesis, especially pronounced at the earlier stages of the reaction.

| Entry | PS/IE, mol/mol | Conv. PS, % | Conv. IE, % | Yield (IE-IE/PS/PS/IE-PS), % | Observed ratio (IE-PS/PS-PS) | Predicted ratio for random coupling (IE-PS/PS-PS) |
|-------|----------------|-------------|-------------|------------------------------|------------------------------|---------------------------------------------------|
| 1 | 1 | 25 | 30 | 6/4/17 | 4.5 | 2 |
| 2 | 1 | 63 | 78 | 22/14/34 | 2.4 | 2 |
| 3 | 1 | 85 | 94 | 24/19/46 | 2.4 | 2 |
| 4 | 0.66 | 93 | 93 | 15/15/63 | 4.3 | 3 |
| 5 | 1.56 | 95 | >95 | 30/27/40 | 1.5 | 1.3 |
| 6 | 3.56 | 20 | 35 | 15/7/6 | 0.78 | 0.56 |
| 7 | 3.56 | 47 | 43 | 15/17/13 | 0.77 | 0.56 |

Reaction conditions: HGII (0.01 mol%), RT, N₂.

## Polyesters based on IE-IE (P-IE-IE), and IE-IE, IE-PS and PS-PS dimers (P-PS-IE)

As a model reaction to convert the obtained bisphenols into final products polyesterification with succinyl chloride was chosen. While P-IE-IE was reported before[26], (P-PS-IE) was not previously described. Figure 7a and Supplementary Fig. 28–29 show ¹H and ¹³C NMR spectra of P-PS-IE in DMSO-d6. The PS/IE ratio (PS/IE = 1) in the starting mixture and in the final polymer stayed largely unchanged (Supplementary Fig. 13). This indicates that the reactivity of all three dimers towards polycondensation was similar, and the final ratio of PS and IE units can be predicted from the initial composition of the mixture.

Thermal properties of the polymers were assessed by differential scanning calorimetry (DSC) and thermogravimetric analysis (TGA) (Fig. 7b). The main drastic difference between P-IE-IE and P-PS-IE is the absence of crystallization peak in case of P-PS-IE, where only a clear glass transition occurs at 149.2 ± 0.7 °C (mid.point, Fig. 7c). In case of P-IE-IE there are three transitions: glass transition (95.3 ± 0.6 °C, mid.-point; Fig. 7c), crystallization (186.2 ± 0.9 °C) and melting (232.9 ± 0.6 °C). This is expected from a more regular structure of P-IE-IE vs. P-PS-IE allowing for a better packing of polymer chains. P-PS-IE exhibited ca. 50 °C higher $T_g$, reflecting impeded mobility of the polymer chain due to an additional methoxy group in the aromatic ring of PS fragment. Both polymers exhibited two mass losses processes. P-PS-IE polymers demonstrated superior thermostability with $Td_{5\%}$ being around 40 °C higher than for P-IE-IE (304 ± 3 °C vs. 263.4 ± 0.5 °C). The thermal properties of P-IE-IE are comparable (or even superior) to

commercial polyesters (PET, $T_g$ = 80 °C, $T_m$ = 251 °C)[26]. The thermal properties of P-PS-IE, which exhibited high $T_g$ = 149.2 ± 0.7 °C are matching commercial amorphous BPA-based polycarbonates ($T_g$ = 150 °C)[48].

Molecular weight determination of the prepared polymers was hindered by their poor solubility in THF and DMF−solvents generally used for gel permeation chromatography (GPC). Moreover, while the polymers are soluble in DMSO, polystyrene standards needed for analysis show a poor solubility in DMSO at room temperature. These factors made the estimation of the molecular weight of the polyesters challenging and therefore, the molecular weights were estimated via Diffusion-Ordered NMR Spectroscopy (DOSY). The experiments were performed in DMSO-d6 at 100 °C (Supplementary Fig. 14). $M_W$ were estimated to be 13041 and 11450 Da for P-IE-IE and P-PS-IE respectively. However, both samples exhibited a high polydispersity expected for step-growth polymerization.

Successful polymerization of the mixture consisting of PS-PS, PS-IE and IE-IE dimers suggests that polymerization of dimers obtained from lignin-derived oil is feasible. Moreover, it allows for potential additional tuning of the final properties upon adjustment of the initial PS/IE ratio.

## Preparation of lignin-derived oil enriched in PS and IE

After getting a better understanding of the system using model mixtures of PS and IE, we focused on transferring the developed protocols on a lignin oil derived directly from wood. While an optimization of the reaction conditions for hydrogen-free RCF is outside the scope of the current study and can be found elsewhere[24], in this work the relationships between the composition of the reaction mixture and its reactivity were explored. Specifically, a particular focus was placed on the effect of allylic species, as they were found to exert most significant effect on the catalytic conversion of target IE and PS. The screening was limited to the reaction conditions for the specific set-up used in this work (Supplementary Table 1). RCF was performed using birch sawdust (particle size < 0.3 mm). Qualitative analysis of the lignin oil (LO) revealed that performing RCF at lower temperature (190 °C) results in a mixture containing a significant amount of sinapyl and coniferyl alcohol and their derivatives, which is in a good agreement with the mechanistic proposal for lignin depolymerization established in previous studies (Supplementary Fig. 15)[39,40]. LO obtained at 210 °C already contained an increased amount of products of overreduction (e.g. propyl and ethyl syringol, Supplementary Fig. 16). For this set-up the amount of allylic species can be minimized, while the yield of PS and IE can be maximized by performing the RCF reaction at 200 °C for 4 h with a slow preheating from room temperature to 200 °C (Supplementary Figs. 17– 18). However, when sawdust with large particle size (non-sieved wood) was used for RCF at 200 °C, the obtained LO still contained minor amounts of sinapyl alcohol (Supplementary Fig. 33).

A separation of monomers from lignin dimers and oligomers can be accomplished either via a distillation or via a solvent extraction. Due to a simplicity of the protocol on the laboratory scale, an isolation of monomers from the LO via an extraction with hexane (reflux, overnight) was performed, previously reported by the Sels research group[20]. In addition, extraction can help to minimize the amount of oxygenated species (including allylic alcohols) in the lignin monomers oil (LMO), due to their higher polarity and as such a lower solubility in hexane. At the same time purification of LMO from allylic species via distillation is challenging due to close b.p. of coniferyl alcohol−340 °C and PS−353 °C.

In this work, two types of LO and their corresponding lignin monomers oils (LMO) were studied: obtained via RCF at 190 °C (LO-190, LMO-190) and at 200 °C (LO-200, LMO-200). LO-200 showed a minor presence of allylic species (sinapyl alcohol), while in case LO-190 both sinapyl alcohol and coniferyl alcohol were present in high

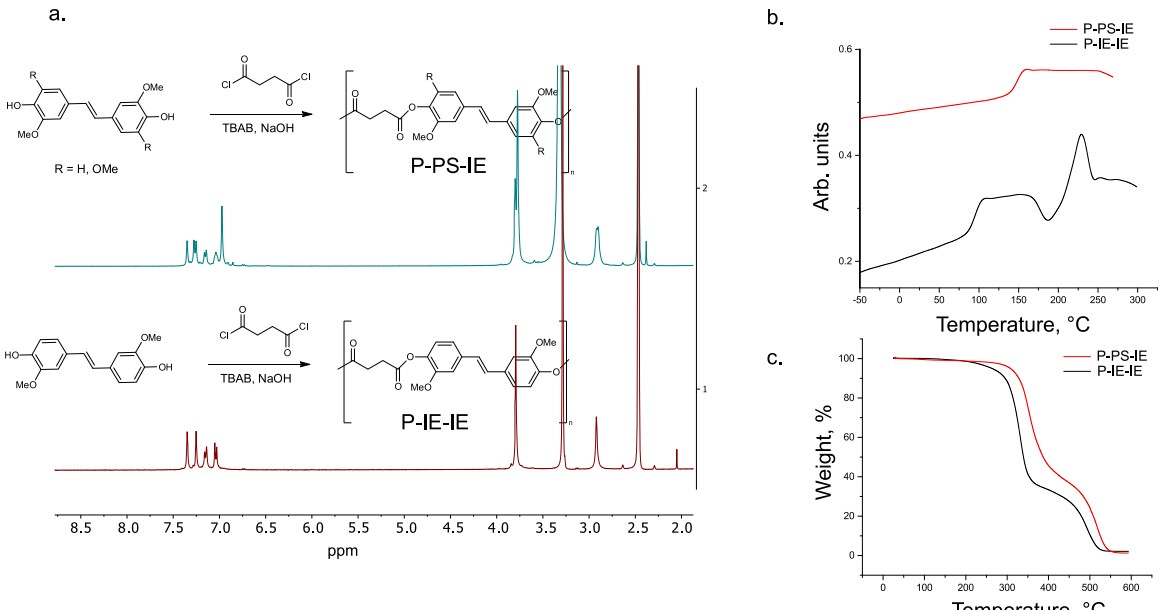

**Fig. 7 | Structural and thermal properties of P-IE-IE and P-PS-IE. a** [1]H NMR spectra of P-IE-IE and P-PS-IE in DMSO-d6. **b** Representative differential scanning calorimetry (DSC) curves of P-IE-IE and P-PS-IE. P-IE-IE exhibits three clear thermal transitions (glass transition, crystallization and melting), while P-PS-IE demonstrated only glass transition. This can be rationalized by a more regular structure of P-IE-IE compared to P-PS-IE allowing for its packing into crystals. **c** Representative thermogravimetric analysis (TGA) graphs (under $N_2$) of P-IE-IE and P-PS-IE. TGA curves indicate two mass loss processes.

amounts. An extraction with hexane (reflux, overnight) allowed to almost fully remove allylic species from LMO-200 and significantly reduce their amount in case of LMO-190 (Supplementary Fig. 19-22). However, sinapyl and coniferyl alcohols were still clearly present in LMO-190. Total content of IE and PS in LMO was determined by GC-FID (Supplementary Fig. 23) and [1]H NMR (Supplementary Fig. 24). Both methods provided similar quantitative results. A total content of PS and IE was found to be 67-72 wt% (LMO-200) and 55-57 wt% (LMO-190) with a molar ratio of PS/IE of 2.3 (LMO-200) and 2 (LMO-190). A general pathway to LMO is schematically presented on Fig. 8a. [1]H NMR of LMO-200 is shown on Fig. 8b. Even though the mixture consists of multiple components, signals corresponding to PS and IE are clearly distinguishable, which allows monitoring the transformation of PS and IE via [1]H NMR (Fig. 8c). A full description of the identified compounds in LMO-200 are given in Supplementary Table 2.

## Metathesis of lignin-derived oil

Upon obtaining the LMOs, their performance in metathesis were studied by performing the reactions under standard reaction conditions (room temperature, inert atmosphere (glovebox), HGII), and monitoring the progress of the reaction via conversion of PS (Fig. 8c). LMO-190 was tested first, where both coniferyl and sinapyl alcohols were detected. When the reaction was performed with 0.07 mol% of HGII the conversion of PS reached 21% after 22 h (Table in Fig. 8d, entry 1). Increasing the catalyst loading to 0.14 mol% resulted in improved, but still moderate conversion (50%, Table in Fig. 8d, entry 2). LMO-200 demonstrated a significantly better performance, where the conversion of PS reached 50% already with 0.07 mol% HGII and 79% with 0.14 mol% HGII (Table in Fig. 8d, entry 3–4). The conversion of PS in LMO-200 dropped only slightly when catalyst loading was lowered to 0.11 mol% (75 %, table in Fig. 8d, entry 6). These results are in good agreement with the studies performed on model compounds, which revealed that the presence of allylic species (even in minor amounts) exhibits a clear deleterious effect on the metathesis of IE.

The developed protocol for in situ conversion of allylic alcohols was probed on LMO, obtained from non-sieved wood (RCF at 200 °C, Supplementary Fig. 33) containing a minor amount of sinapyl alcohol.

Using 0.07 mol% of HGII led to 31% conversion of PS. When the reaction was performed in a two-step manner (90 °C, HGII 0.025 mol%, 14 h, followed by the reaction at room temperature with 0.07 mol% of HGII, 22 h), the conversion improved to 77%. Note, that while the total amount of HGII used was 0.095 mol%, the product was not formed during the pretreatment step. This result indicates that the developed tandem protocol can be applied to mixtures derived directly from wood, however, additional studies will be required to identify all the transformations occurring during the pretreatment step.

Then we tested if an enhanced reactivity of PS in the presence of IE, discovered during model studies, is also observed in case of metathesis performed on lignin-derived mixture. IE was added to the reaction mixture in 60 wt% (50 mol%). Indeed, an addition of IE allowed to lower the catalyst loading more than twice resulting in 83% and 89% conversion of PS respectively with 0.055 mol% HGII and 0.065 mol% of HGII (relative to total amount of PS and IE) (Table in Fig. 8d, entries 7-8). Note, that the reaction was evaluated based on the conversion of PS, which solely comes from LMO, rather than on the conversion of IE, which was added to the reaction mixture in its pure form.

A gaseous phase of the reaction mixture was also analyzed using head-space GC-MS method (Supplementary Fig. 25). Expectedly, in addition to toluene (used as a solvent) the chromatogram revealed the presence of butene as the only product.

Overall, the results obtained from lignin-derived mixtures (LMO-190 and LMO-200) are in line with model studies signifying the importance of the elimination of allylic alcohols from the reaction mixture and an enhanced reactivity of PS in the presence of higher amount of IE.

## Isolation and photo-chemical properties of lignin derived dimers

One of the key advantages of the system is a simplicity of the isolation of the final products. Final dimers (PS-PS, PS-IE and IE-IE) possess a significantly higher polarity and boiling points compared to both unreacted starting PS and IE and other monomeric impurities, which offers an easy purification either via distillation or extraction with non-polar solvents. The obtained dimers in their

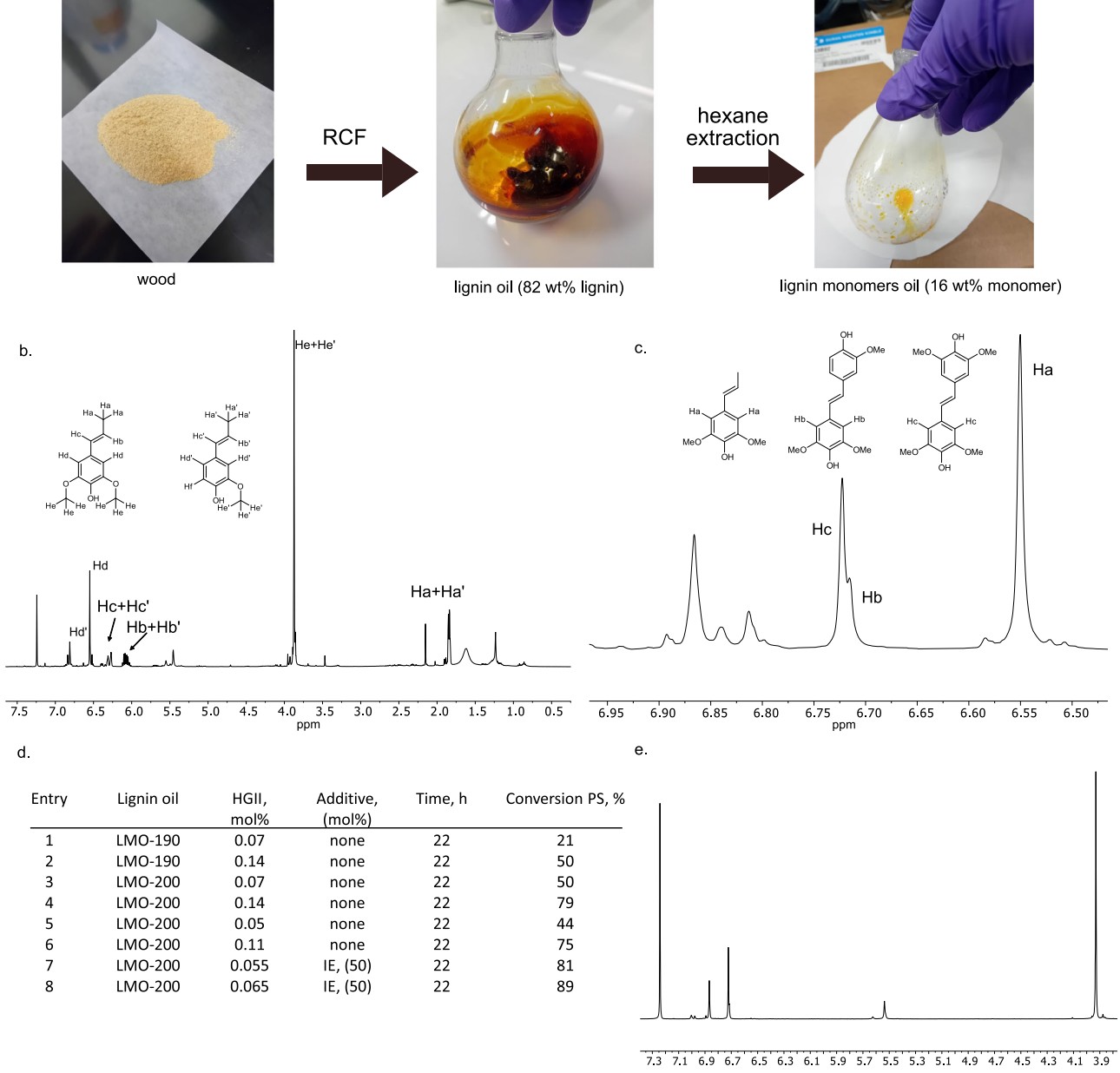

**Fig. 8 | Olefin metathesis of lignin monomers oil (LMO-190 and LMO-200) using Hoveyda Grubbs 2nd generation catalyst (HGII). a** The transformation pathway from wood to LMO: reductive catalytic fractionation (RCF) of birch sawdust, hexane extraction of lignin monomers from lignin oil (reflux, 16 h). **b** [1]H NMR of LMO-200 enriched in propenyl syringol (PS) and iso-eugenol (IE). Despite the complexity of the mixture the signals of PS and IE are clearly distinguishable, which allows monitoring the transformation by [1]H NMR. **c** An example of the reaction mixture of the metathesis of LMO. Characteristic peaks corresponding to PS, PS-PS, PS-IE dimers. **d** Optimization of olefin metathesis of LMO. LMO-200 with minimized amount of allylic alcohols required significantly lower catalyst loading compared to LMO-190 containing higher amounts of sinapyl and coniferyl alcohols. **e** An example of [1]H NMR spectrum of PS-enriched purified lignin dimers (PS-PS, PS-IE). A simple wash of the reaction mixture with $Et_2O$ allowed to obtain dimers of high purity.

pure form were isolated via two dispersion/centrifugation cycles using $Et_2O$ as a solvent (Fig. 8e).

The presence of stilbene moiety in the structure of the dimers renders them photo-chemically active. Several transformations can occur upon their exposure to UV-light, including [2 + 2] cycloaddition, cis-trans isomerization, Mallory reaction, etc. This reactivity can be beneficial allowing for the diversification of the products, however, it can also compromise the stability of the reaction mixtures. To this end, a targeted study of the stability of the dimers upon the exposure to UV light (365 nm) in $CDCl_3$ and $CD_3OD$ was performed. In $CDCl_3$ upon exposure to UV light (365 nm) the initially yellow solution rapidly (within minutes) acquires a red color, which changes back to yellow

upon exposure to white light (Supplementary Fig. 26a), indicating a photochromic reaction. However, [1]H NMR of the reaction mixture revealed other irreversible and non-selective transformations (Supplementary Fig. 26b). Thus, the solutions of the dimers in $CDCl_3$ need to be handled with caution in the presence of UV light. The dimers dissolved in $CD_3OD$ were stable upon exposure to UV light. No color change was noted and the [1]H NMR spectrum revealed a clear cis-trans isomerization (Supplementary Fig. 27).

**Polymerization of lignin-derived dimers into polyesters**
As a final part of this work, the preparation of polyesters from lignin-derived mixtures of PS-enriched dimers was explored. PS-enriched

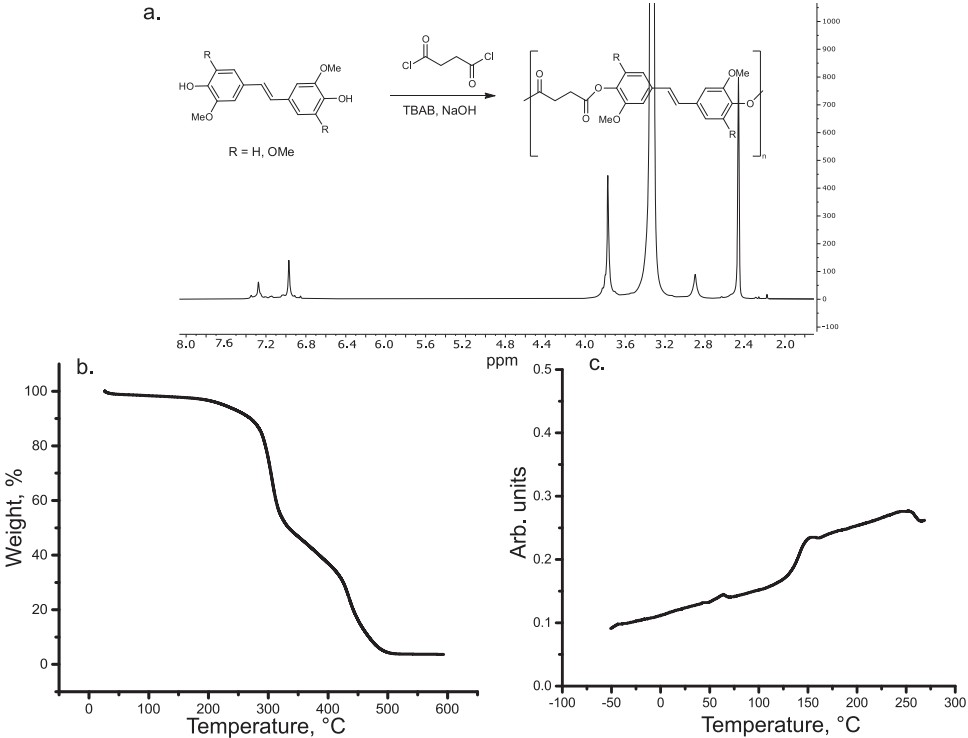

**Fig. 9 | Structural and thermal properties of polyester prepared from lignin-derived mixture of dimers. a** $^1$H NMR spectrum of the polyester. **b** A representative termogravimetric analysis (TGA) graph of the polyester (N$_2$ atmosphere). The curve exhibits two major mass loss events. **c** A representative differential scanning calorimetry (DSC) curve of the polyester indicating only one phase transition - glass transition.

mixtures are more challenging with regards to the polymerization. This is due to higher steric demand of PS fragments compared to IE and higher pKa of the phenolic groups, which can slow down their nucleophilic attack on the succinyl chloride. In addition, due to a highly electron-rich nature of PS units they are especially prone towards oxidation under basic conditions. To probe if lignin-derived mixtures of dimers with a high PS content still can be successfully polymerized, a mixture with PS/IE ratio of 6.65 was chosen (Supplementary Fig. 30). PS/IE ratios can be easily tuned by an addition of IE or by choosing a different source of wood (e.g. soft wood). The polyester was prepared following the same procedure as in case of model compounds (Fig. 9a) via polycondensation with succinyl chloride. It is important to underline that a second product of the metathesis reaction – butene-2, is a precursor for the production of succinyl chloride via established synthetic routes (via maleic anhydride)[49,50]. Even though this transformation is outside the scope of the current study the final polymeric materials can thus be considered all-carbon biobased. $^1$H NMR of the final polymer is shown on Fig. 9a (Supplementary Fig. 31), $^{13}$C NMR is shown on Supplementary Fig. 32. The thermal properties of the polymer using DSC and TGA (Fig. 9b, c) were evaluated. $T_g$, $Td_{5\%}$ and $Td_{50\%}$ were found to be 140.3 ± 0.7 °C, 231 ± 6°C and 330 ± 4 °C respectively. Mw of the polymer (16330 Da) was estimated using DOSY NMR and is in a similar range as Mw of P-PS-IE and P-IE-IE (13041 and 11450 Da). This part of the work demonstrates a final step of the proposed lignin monomers valorization strategy, which addresses the entire process from wood to final polymeric materials.

## Discussion

In summary, an alternative solution to tackle a high complexity of lignin depolymerization oil, neither requiring a purification of the mixture nor a high selectivity of the depolymerization step, is reported. Common approaches relying on catalytic funneling, essentially converting the mixture into a single product, necessitate harsh reaction conditions and cause the loss of valuable functionality of lignin monomers. In addition, these methods generally require high selectivity during lignin depolymerization step as well. Instead, a mild pathway allowing a selective conversion of target IE and PS into easily separable bisphenols and butene-2 via olefin metathesis is demonstrated. Rather than purifying the lignin depolymerization oil, its composition is optimized. The main species responsible for catalyst poisoning, allyl alcohols, are identified and their presence is minimized in the final mixtures. Further, a protocol for their in situ conversion is also developed. In addition, the PS/IE ratio is optimized to increase the reactivity of PS, which allowed up to 89% conversion of PS at room temperature with HGII loading as low as 0.065 mol%. With the current experimental protocol, the reusability of the catalyst appears cumbersome. In a future effort a heterogeneous version of the catalyst can be considered[51]. In this case mass transfer limitations will need to be addressed, due to a high viscosity of the reaction mixture at higher conversion levels. While thorough toxicological studies of the prepared bisphenols are outside the scope of this work, the literature reports indicate a significantly reduced estrogenicity of IE-IE compared to BPA[27]. It is worth noting that the production of bio-based BPA analogs was a subject of previous studies[31,52,53]. Some approaches offer attractive transition metal-free pathways via e.g. zeolite-catalyzed alkylation of guaiacol with IE[52] or ethyl guaiacol dimerization using formaldehyde[53]. However, since those methods rely on the reactivity of an aromatic ring, the selectivity of the processes is more difficult to control, which complicates a direct application of the processes to lignin oils. In contrast, the presented herein methodology is selective towards olefins and as such allows for the selective formation of the target bisphenols in the presence of other phenolic compounds.

Obtained bisphenols have a high potential to serve as substitutes for BPA. To demonstrate their application in production of polymers, we prepared a polyester from lignin depolymerization oil (enriched in PS unit) and succinyl chloride. It is important to mention that a second product of metathesis step, butene-2, is a known precursor for maleic and succinic acids. Although this transformation is outside of the

scope of the current study, the final polyesters can be considered as fully lignin-based (for carbon flow analysis see Supplementary Fig. 36). The polyester demonstrated high glass transition temperature ($T_g = 140.3\,°C$) and thermal stability ($Td_{50\%} = 330\,°C$). Transition from the laboratory scale to pilot and eventually industrial scale will require reconsideration of several process steps. For the separation of the lignin monomers from high molecular weight fragments distillation might be a preferential approach (vs. currently implemented hexane extraction). In addition, polyesterification protocol will need further attention, where the use of succinyl chloride and dichloromethane for interfacial polymerization is avoided. With this regard, life cycle assessment (LCA) studies are pivotal.

Overall, this study addresses major challenges in the pursuit of economically and sustainably generating value-added products from complex bio-derived mixtures—separation of the products, selectivity and catalyst poisoning. These findings have the potential to extend beyond lignin-derived mixtures for application to other types of biomass streams. Optimization of the catalyst structure for higher tolerance to these impurities can further improve the efficiency of the process. The generated bisphenols PS-PS, PS-IE, and IE-IE are highly functional and can be considered as platform molecules enabling their further diversification. Development of such facile diversification methods is of interest for future studies and would allow for building of a library of the bisphenols to enable a rapid screening of their toxicity as well as material properties of the resultant polyesters. In addition, polymers based on PS-PS, PS-IE, and IE-IE can find high-end applications in photo-sensitive materials operating via cis-trans isomerization and [2 + 2] cycloaddition reactions.

## Methods

### General information

Unless otherwise stated all chemicals, solvents and materials were purchased from Sigma Aldrich (Merck). Sinapyl alcohol and coniferyl alcohol were purchased from Cayman Chemical. UltraNitroCat (UNC) was purchased from Strem Chemicals. Pt100 RTD Probe was purchased from Evolution Sensors and Controls LLC. A digital PID temperature controller was purchased from BriskHeat. Quick-Install Fiberglass Pipe Insulation Tube was purchased from MCMASTER-CARR. Birch sawdust (mix of Betula Pendula and Betula Pubescens) was provided by Vanhälls Såg AB.

GC-MS analyses were carried out on a PerkinElmer Clarus 580 with connected SQ8S mass spectrometer (Waltham, MA) using an Elite-5-MS (60 m × 0.25 mm i. d. × 0.25 μm) capillary column (PerkinElmer). GC-FID analyses were carried out on a PerkinElmer Clarus 580 with connected flame ionization detector using an Elite-5-MS (60 m × 0.25 mm i. d. × 0.25 μm) capillary column (PerkinElmer).

Gaseous products of the reaction mixtures were analyzed using Shimadzu Scientific Instruments Triple-Quadrupole 8050 GC-MS NX system, equipped with an AOC-6000 autosampler, a GC-2030 column oven and a GCMS-8050 mass spectrometer. Headspace sampling was performed with the autosampler tool of the AOC-6000. Samples were equilibrated for 1 min at 40 °C, the syringe was held at 60 °C. 100 μL of sample volume was injected with a split injection (split ratio 50) into the injection port of the GC which was held at 250 °C. Helium was used as a carrier gas with a column flow rate of 0.5 mL/min. The sample was separated on a Shimadzu SH-Rxi-5Sil MS column (Length 30 m, 0.25 mm ID, 0.25 μm film thickness) with the following temperature program: 40 °C (hold 2 min) to 200 °C with a heat rate of 10 °C /min (hold 2 min). The MS interface and ion source were held at 250 °C, sample ionization was performed with electron ionization (70 eV) and positive ions were analyzed with a Q3 scan between 10 and 400 m/z (event time 100 ms) for the full time of the GC run (20 min). Measurements and data post-processing were performed with GCMS Real Time Analysis and Postrun Analysis 4.50 SP1.

Mass spectrometric measurements were performed with a Thermo Fisher QExactive Orbitrap MS system using continuous injection with a syringe. Samples were prepared in a glove box and loaded into a gas tight syringe, Hamilton 1750, for sample injection. Samples were held at room temperature and continuously injected using a syringe pump at 10 μL/min. Electrospray was used for desolvatization and ionization, with the electrospray needle held at +3.5 kV. Compressed air was used as desolvation gas, capillary temperature was at 320 °C, probe heater temperature at 40 °C and sheath gas flow was at 5 L/min. Resolution was set to at least 35,000 M/ΔM. Mass spectra were recorded in the range of 150 to 2000 m/z in positive ion mode and in smaller mass windows to increase relative intensity of low intensity ions. Measurements and data post-processing were performed with Thermo Xcalibur 4.1.31.9.

TGA measurements were performed using TGA 550 (TA instruments) Discovery series. DSC measurements were performed using DSC 250 (TA instruments) Discovery series.

$^{1}$H and $^{13}$C NMR spectra were recorded on either a 400, 500, or 600 MHz Bruker NMR instrument.

### General reaction procedure for self- and cross- olefin metathesis of IE and PS

In a glovebox a corresponding amount of substrate (IE and/or PS, 60–80 mg, 0.3–0.5 mmol) and anhydrous toluene (1 mL, for the reactions performed in solution) were placed in a dry vial equipped with a dry stir bar. A corresponding amount of a catalyst (Hoveyda-Grubbs Catalyst 2$^{nd}$ generation (HGII), UltraNitroCat (UNC) or nitro Grela catalyst (nG)) was added as a solution in toluene (10–15 mg/5 mL, 0.003–0.005 M). The reaction mixture was left to react under stirring for a corresponding period of time. As the reaction progresses the reaction mixture solidifies. The reaction was quenched with ethyl vinyl ether and analyzed by $^{1}$H NMR.

### General reaction procedure for olefin metathesis of IE in the presence of additives

In a glovebox a corresponding amount of IE (60–80 mg, 0.36-0.5 mmol) was placed in a dry vial equipped with a dry stir bar. A corresponding amount of an additive (vanillic acid, sinapyl alcohol) or a solution of an additive in toluene (for toluene soluble additives: vanillin, furfural, 3-Phenyl propanol, cinnamyl aldehyde, cinnamyl alcohol, methyl cinnamyl ether, allyl alcohol, cinnamyl acetate) was added to the vial. The reaction mixture was stirred for several minutes. After that a corresponding amount of a catalyst (HGII) as a solution in toluene (10–15 mg/5 mL, 0.003–0.005 M) was added. The reaction was quenched with ethyl vinyl ether and analyzed by $^{1}$H NMR. In order to isolate an effect of additives from other factors affecting the conversion a reference reaction under the same reaction conditions in the absence of any additives was performed in parallel for each run.

### Preparation of the reaction mixtures for $^{1}$H NMR studies of the decomposition of the catalyst (HGII)

In a glovebox to a corresponding amount of a substrate: cinnamyl alcohol (12.9 mg, 0.096 mmol), cinnamyl methyl ether (12 mg, 0.081 mmol) or cinnamyl aldehyde (8.6 mg, 0.064 mmol), was placed in a dry vial equipped with a dry stir bar, a solution of HGII in toluene-d8 was added to reach 4 mol% catalyst loading relative to the substrate followed by an addition of a corresponding amount of toluene-d8 to reach 0.14 M concentration of a substrate. The reaction mixtures were left to react for a corresponding period of time. To monitor the reactions, dried NMR tubes and septa were brought in the glovebox. NMR tubes were filled with the corresponding solution, sealed with septa and parafilm tape.

When the decomposition of the catalyst was studied at 110 °C, the reactions were performed in a similar fashion, but the reaction vessels were placed in a preheated sand bath for a corresponding time.

General procedure for the tandem cinnamyl alcohol isomerization/IE self-metathesis reaction:

In a glovebox into a dry vial equipped with a dry stir bar a corresponding amount of IE (200-300 mg, 1.22 -1.83 mmol) and a solution of cinnamyl alcohol in toluene (5 mol% relative to IE) were placed. The reaction mixture was stirred for a few minutes. After that a corresponding amount of HGII (0.025 mol%) as a solution in toluee or carbonylchlorohydridotris(triphenylphosphine)ruthenium(II), RuH 0.027 mol% (as solutions in toluene and dichloromethane, 1/1) were added. The vial was sealed and transferred into a sand bath preheated to 90 °C. The reaction progress was monitored by GC-MS. When cinnamyl alcohol was almost fully consumed (12–24 h), the reaction mixture was cooled down to room temperature and a second portion of the catalyst (HGII) 0.02-0.025 mol% was added as a solution in toluene. The reaction mixture was left to react for 1 h and analyzed by [1]H NMR.

### General procedure for the preparation P-IE-IE and P-PS-IE
A corresponding amount of IE-IE dimer (96.1 mg, 0.35 mmol) or a mixture of IE-IE, PS-PS and IE-PS (255 mg, 0.848 mmol) dimers were placed in a flask equipped with a stir bar followed by an addition of tetrabutylammonium bromide (3.8 mg, 0.012 mmol (for P-IE-IE) and 8.8. mg, 0.027 mmol (for P-PS-IE)). The flask was sealed with a septum and the atmosphere in the flask was exchanged to argon. A corresponding amount of a solution of 0.2 M NaOH (5 mL for P-IE-IE and 15 mL for the mixture of P-PS-IE) was added and the reaction mixture was left to stir until a sufficient solubility of the starting materials was achieved. For the mixtures containing PS units it is very important to avoid the presence of oxygen due to a possible oxidation of PS units upon the deprotonation. A corresponding amount of succinyl chloride, dissolved in DCM was added (54.6 mg, 0.35 mmol (for P-IE-IE) and 131.5 mg, 0.848 mmol (for P-PS-IE)). Upon the addition of succinyl chloride a precipitate begins to form. The reaction mixture was left to react for 24 h. Upon the completion the reaction mixture was acidified to $pH = 1$ with aqueous HCl. The precipitate was washed with water, methanol and acetone to give final products. The polymers were dried under vacuum and analyzed by NMR. For P-IE-IE NMR data match to previously reported (Supplementary Fig.37)[26].

P-PS-IE (170 mg, 52%):

[1]H NMR (400 MHz, DMSO-d6): δ 2.96 (bs), 3.83 (s), 3.81 (s), 7.01 (m), 7.08 (m), 7.18 (m), 7.30 (m), 7.39 (m).

[13]C NMR (151 MHz, DMSO-d6): δ 170.6, 170.4, 170.2, 170.1, 152.3, 151.5, 148.3, 139.2, 136.5, 135.90, 129.1, 129.0, 128.7, 128.6, 128.1, 128.0, 123.4, 119.5, 110.8, 110.7, 103.8, 103.7, 56.5, 56.3, 29.0, 28.7, 28.6.

### General procedure for reductive catalytic fractionation (RCF) of wood
Birch sawdust was sieved through a sifter (Mesh-Hole Size 0.3 mm). The sieved sawdust (2.6 g) and 0.260 gram of Pd/C (5 wt% Pd loading, dry basis) was added into a teflon reaction vessel equipped with a stir bar followed by the addition of 40 mL of ethanol and 40 mL of water. The reaction vessel was placed in a metal reactor and sealed. A Pt100 RTD Probe was attached to the body of the reactor using a hose clamp. A heating band was wrapped around the reactor and covered with the fiberglass insulation. The heating was performed using a digital PID temperature controller (briskheat). The reactor was left to reach a desired temperature (190 °C, 200 °C or 210 °C) followed by the reaction at the final temperature for 4 h (for the heating profile see Supplementary Fig. 18). Upon the completion of the reaction the reactor was cooled down using cold water. The reaction mixture was filtered to remove the catalyst and the solid residue (mainly cellulose). The filtrate was concentrated under vacuum. Lignin oil (LO) was isolated via three-fold extraction with EtOAc, washed with brine and dried over $Na_2SO_4$ followed by the evaporation of EtOAc. LO appears as a brown oil or sticky brown solid. LO was subjected to silylation via the

following procedure: 5–10 mg of LO was dissolved in 1 mL of anhydrous THF in GC vial, 50 μL of anhydrous pyridine and 100 μL of N,O-Bis(trimethylsilyl)trifluoroacetamide were added into the vial. The vial was placed in a sand bath preheated to 50 °C for 15 min. The silylated sample of LO was analyzed by GC-MS.

### General procedure for the isolation of lignin monomers oil (LMO) via hexane extraction
A flask containing LO (1.5–2.5 g) and a stir bar was filled with hexane (90–150 mL). A reflux condenser equipped with a septum connected to inert gas via a needle was connected to the flask. The reaction mixture was left under reflux overnight (16 h). Upon the completion of the extraction the reaction mixture was filtered through a paper filter. Upon the extraction of monomers, LO becomes solid and stays on the walls of the flask. The filtrate (light yellow liquid) was subjected to vacuum evaporation to give a final LMO. LMO was further dried under high vacuum overnight.

### General procedure for olefin metathesis of LMO
LMO with a known amount of PS and IE was dissolved in anhydrous toluene (100–200 mg in 1–2 mL). In a glove box an aliquot of the solution was transferred into a dried vial equipped with a dried stir bar. A corresponding amount of HGII was added as a solution in toluene. The reaction was left under stirring in the glovebox for a corresponding amount of time. The progress of the reaction was monitored by [1]H NMR.

### Isolation of PS-PS, PS-IE and IE-IE dimers from the olefin metathesis reaction mixture
Upon the completion of the metathesis reaction of LMO, toluene was removed from the reaction mixture under reduced vacuum. The reaction mixture (69 mg) which appears as a white to yellow solid (or sticky solid) was dispersed in $Et_2O$ (2 mL) and subjected to centrifugation. The supernatant was removed, and the solid residue was re-dispersed in 1 mL of $Et_2O$ and subjected to centrifugation again. A solid residue (white solid) was dried under high vacuum to give a final mixture of dimers (26.6 mg, 38.5 wt%).

### Polymerization of mixture of dimers obtained from LMO
The polycondensation of LMO-derived mixture of dimers was performed according to the general procedure. 51.3 mg (0.158 mmol) of the mixture of dimers (PS/IE = 6.55/1, Supplementary Fig. 30) and TBAB (1.6 mg, 0.05 mmol) was dissolved in 3 mL of 0.2 M NaOH under inert atmosphere. 26 mg (0.167 mmol) of succinyl chloride dissolved in 4 mL of DCM was added to the reaction mixture. The reaction was left to proceed for 24 h. Upon the completion of the reaction, the mixture was acidified to pH = 1 with HCl. The precipitate was washed with water, methanol and acetone to give a final product (41.1 mg, 64%).

[1]H NMR (400 MHz, DMSO-d6): δ 2.93 (bs), 3.81(bs), 7.01 (bs), 7.31 (bs), 6.89-7.39 (m)

[13]C NMR (151 MHz, DMSO-d6): δ 170.1, 152.3, 135.9, 129.1, 128.1, 103.73, 6.5, 28.6.

### Estimation of Mw of polymers via DOSY [1]H NMR experiment
The determination of the Mw of the prepared polyesters was carried out via Diffusion-Ordered NMR Spectroscopy (DOSY). The calibration curve was built using polystyrene standards (Mw = 2500, 5000, 9000 and 30000). Due to a poor solubility of PS in DMSO-d6 at room temperature the experiments were run at 100 °C. To minimize the effect of convection NMR tubes of smaller diameter (3 mm) were used and a pulse sequence with convection compensation was implemented. DOSY NMR experiments were performed using 500 MHz Bruker NMR instrument with the following parameters: pulse sequence– Dbppste_cc; relaxation delay–1 s; number of increments–15; Lowest gradient strength–1130, highest gradient strength–28261; diffusion

gradient length−2 ms; diffusion delay−50 ms. The spectra were analyzed using Mestrenova software. For polystyrene standards a peak at 6.75–7.25 was used to build diffusion decay curves, for the analyzed polyesters–spectra were integrated in the whole region 3.5–7-5 ppm (Supplementary Fig. 14). Diffusion coefficients were obtained using either mono-exponential or three parameter exponential fits.

## Differential scanning calorimetry (DSC) measurements
DSC measurements were performed on DSC 250 (TA instruments) Discovery series. 1.5–2 mg of the sample was placed in an aluminum pan and sealed with an aluminum lid. Temperature profile: −75 °C–270 °C (for P-PS-IE and polymer based on lignin-derived dimers) or to 300 °C (for P-IE-IE), ramp: 10 °C/min; 270 °C − (−75 °C), ramp: 10 °C/min (2 cycles). The reported values ($T_g$, m.p. and Tc) are calculated from the second heating cycle.

## Thermogravimetric analysis (TGA)
TGA measurements were performed using TGA 550 (TA instruments) Discovery series. 1.5-2 mg of the sample was placed in a Pt pan. Temperature profile: 20 °C – 600 °C, ramp: 10 °C/min. $N_2$ atmosphere.

## Data availability
The datasets generated and/or analyzed during the current study are supplied in the supplementary information. If additional data or information is sought, this will be provided by the corresponding authors upon request.

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

## Acknowledgements

This research made use of the Chemical and Biophysical Instrumentation Center at Yale University (RRID:SCR_021738). The authors acknowledge the use and research support provided by the Yale Mechanical and Thermal Analysis Instrumentation Core for the DSC and TGA experiments. Equipment was purchased with funds from Yale University and the NSF MRI grant CHE-1828190. This project has received funding from the European Union's Horizon 2020 research and innovation program under the Marie Skłodowska-Curie grant agreement No 101023166-SE (E.S.).

## Author contributions

E.S. conceptualized the idea, performed most of the experimental work, and wrote the manuscript and the supplementary information. L.R.S. performed part of the olefin metathesis experiments and optimization of the reductive catalytic fractionation of wood. J.Z. participated in the discussions and in the manuscript preparation. P.A. provided supervision, participated in project design and management, and in manuscript preparation.

## Competing interests

The authors declare no competing interests.
