## [Peer Review File · Nature Communications]

Room temperature catalytic upgrading of unpurified lignin depolymerization oil into bisphenols and butene-2Reviewers' Comments:

Reviewer #1:

Remarks to the Author:

The manuscript entitled "From wood to polyesters and butene-2: Room temperature catalytic upgrading of unpurified lignin depolymerization oil" submitted by Subbotina et al. describes a value chain from wood, via lignin-derived aryl propenols to yield stilbenes that were demonstrated as BPA substitutes by polymerization with succinyl chloride to yield polyesters.

The work is well executed, and cumbersome side-reaction arising from aryl propenols was identified and to some extent circumvented by using unpolar extractions when using real biomass as substrate and running the reactions at lower temperatures. The generated bisphenols were easily separated by extractions/washes.

The bisphenols were successfully polymerized with succinic acid chloride to generate polyesters with high T_g.

This is a welcomed study, as monophenols generated in lignin-first technologies have been proven difficult to valorize. This study shows a value chain to important building blocks that substitute harmful chemicals derived from fossil resources.

This reviewer's only criticism is the lack of toxicity studies of the stilbene. The authors are encouraged to verify that the generated stilbene has a lower toxicity (estrogen receptor activity), than BPA.

It would be appropriate to include prior art from Sels and Samec on alternative approaches for dimerization: what are the pros/cons?

<https://www.nature.com/articles/s41893-023-01201-w>

<https://pubs-rsc-org.ezp.sub.su.se/en/content/articlehtml/2018/gc/c7gc02989f>

<https://www.sciencedirect.com/science/article/pii/S1359836823001956>

Thus, complementary studies on toxicity and discussion of parent approach compared to prior art would strengthen the manuscript and in this reviewer's view warrant publication in Nature Communications.

Reviewer #2:

Remarks to the Author:

The manuscript describes Lignin an alternative approach is presented, whereby the target monomers from lignin are selectively and directly converted to the final products under mild conditions and easy separation. The method the authors present is based on the hydrogen-free catalytic fractionation of wood to convert lignin into iso-eugenol (IE) and propenylsyringol (PS) enriched oil followed by an olefin metathesis step to yield bisphenols and butene-2, thus, valorising all bio-based carbons. The authors have identified experimentally that the presence of allylic alcohols as major impurities can contribute to catalyst poisoning, even when present in minor amounts. Therefore, they have developed We developed a protocol to solve this problem and improve the catalytic efficiency. The authors have also demonstrated the transformation of bisphenols to polyesters with a high glass transition temperature and good thermal stability.

The following points for the authors to consider:

1. Explain and discuss scientifically the choice of catalysts and the strategy you have chosen, as you

indicate in Figure 2 focusing on the nature of structure of the desired catalyst.

2. Figure 1, clarify which steps you present in the figure you actually study in your present work and add the acronyms in the structures you present in each figure.

3. Have the authors optimise reaction conditions such as amount of catalyst, temperature and concentration of reactants for selecting the best experimental conditions in terms of yield of desired products.

4. The authors have written that Unlike HGII and nG, UNC did not allow for the formation of the product, have they an explanation for the observed result, and if they have investigated the steric factor's role they have written as a possible reason?

5. In Figure 2 it would be useful for the reader to add the yield of the product depending on the choice of the catalyst. Moreover, if there are additional byproducts during the reaction they should discuss in the manuscript.

6. When they study the effect of allylic alcohols on the transformation, figure 2c, the authors have studied their derivatives. Since they indicate that the quantity affects the rate of reaction, it would be useful at least for one derivate to study the effect of concentration, e.g. of cinnamyl methyl ether by varying systematically the concentration from 0 to 30%.

7. The studies of the decomposed catalysts are very important and well-presented and discussed, this is an important part of the manuscript and the presented work.

8. In the section they discuss the "self and cross-metathesis of PS" it would be useful to add the yield of the products in the table you present in Figure 6.

9. A comment if the authors have studied systematically the amount of catalyst for the "self and cross-metathesis of PS" since they have written standard conditions.

10. The authors in the conclusions can comment on the reusability and stability of the catalyst based on their experimental findings, the next steps of investigation, and scale up of the process with possible challenges they envisage in the near future and if LCA studies could be used to strengthen their new methodology.

11. Finally, the work is very interesting and an important topic in the area of utilising lignin-derived mixtures and the authors have developed the methodology for a direct catalytic conversion of wood into polyesters under mild reaction conditions.

Reviewer #3:

Remarks to the Author:

The work by Alena and coworkers proposes and validates an effective and green upgrading protocol for unpurified lignin oil. Briefly, the protocol involves the selective conversion of key components (iso-eugenol (IE) and propenyl syringol (PS)) into bisphenols and butene-2 via olefin metathesis, instead of chemically funneling them into a single product. The notable advantages of this strategy include mild reaction conditions, avoiding high temperatures, and the external need for a hydrogen source. They conducted a solid investigation on: 1. elucidating the key catalyst-poisoning component (allylic species) and the corresponding mechanism; 2. proposing a protocol to tackle this problem; 3. identifying the effect of the PS/IE ratio on the conversion rate. This approach facilitates the efficient isolation of final products and their subsequent polymerization into novel polyesters, demonstrating a sustainable pathway for lignin valorization with the full utilization of bio-sourced carbon. I recommend this paper be published after addressing the following comments:

1. One of the advantages of the proposed strategy is 'all-carbon bio-sourced' and is sustainably/economically favorable. If possible, an analysis of the carbon flow of the lignin-to-polyester process is recommended, considering the broader readership of Nature Communications.

2. In the section 'Self-metathesis of IE and the effect of impurities':

—a. The authors mentioned that 'nG expectedly exhibited faster initiation' without providing theoretical explanations.

- b. The utilization of freshly distilled IE is proven to be more efficient than 'as purchased' IE. Please clearly indicate the purity difference.
- c. The 20 mol% cinnamyl methyl ether lowered the conversion after 1 hour but reached the same level as in the case of pure IE after 11 hours. Please add explanations for the possible reasons.
- d. the reaction mechanism of HGII-mediated PS/IE conversion in SI, if possible, will be helpful to understand the catalysis process and the subsequent HGII decomposition upon allylic species.

3. In the section 'Mechanistic investigation of HGII decomposition by allylic species':

- a. A minor point: positioning the labels of reaction time in Fig. 3a closer to the respective spectra for clearer association.
- b. 'At room temperature, allyl ethers cause the decomposition of HGII at a significantly lower rate than allylic alcohols.' Please add explanations for the possible reasons.
- c. The authors compare RuH and HGII for the tandem sequence. What is the purpose of this comparison? If it solely serves as offering an alternative catalyst while not having a firm conclusion on which one is better, would it be better to put it in SI to have a more coherent streamline in the main discussion?

4. I recommend comparing the properties of lignin-derived polyester and commercially available polyester to further strengthen the practicality of this work.

REVIEWER COMMENTS

Reviewer #1 (Remarks to the Author):

The manuscript entitled "From wood to polyesters and butene-2: Room temperature catalytic upgrading of unpurified lignin depolymerization oil" submitted by Subbotina et al. describes a value chain from wood, via lignin-derived aryl propenols to yield stilbenes that were demonstrated as BPA substitutes by polymerization with succinyl chloride to yield polyesters. The work is well executed, and cumbersome side-reaction arising from aryl propenols was identified and to some extent circumvented by using unpolar extractions when using real biomass as substrate and running the reactions at lower temperatures. The generated bisphenols were easily separated by extractions/washes. The bisphenols were successfully polymerized with succinic acid chloride to generate polyesters with high T_g. This is a welcomed study, as monophenols generated in lignin-first technologies have been proven difficult to valorize. This study shows a value chain to important building blocks that substitute harmful chemicals derived from fossil resources.

This reviewer's only criticism is the lack of toxicity studies of the stilbene. The authors are encouraged to verify that the generated stilbene has a lower toxicity (estrogen receptor activity), than BPA.

We thank the reviewer for this important comment. A thorough toxicological studies of the prepared bisphenols were outside the scope of the current study. However, there are literature reports, where such studies were performed on **IE-IE** dimer and its hydrogenated analogue (**IE-IE red**) (see below).

Thus, in the work by Trita et al. the authors studied the concentration–response curves for hER α (human estrogen receptor alpha) activation by BPA, **IE-IE** and **IE-IE red**. The authors found that the latter two compounds exhibit considerably lower estrogenicity. (<https://pubs.rsc.org/en/content/articlelanding/2017/gc/c7gc00553a>). This work is included in the main text of the manuscript as ref. 27. Evaluation of the toxicity of **PS**-based bisphenols will be a subject of our follow up investigations.

In addition, it is important to mention that the generated bisphenols are considered platform molecules amenable for further diversification. Our follow up studies will focus on the development of a method allowing for a rapid functionalization of the obtained bisphenols to generate libraries of these compounds to enable a facile screening of their toxicological properties. We add the following sentences to the main text of the manuscript:

“While thorough toxicological studies of the prepared bisphenols are outside the scope of this work, the literature reports indicate a significantly reduced estrogenicity of **IE-IE** compared to BPA.”

“...Development of such facile diversification methods would allow the building of a library of the bisphenols to enable a rapid screening of their toxicity as well as material properties of the resultant polyesters.”

It would appropriate to include prior art from Sels and Samec on alternative approaches for dimerization: what are the pros/cons? Thus, complementary studies on toxicity and discussion of parent approach compared to prior art would strengthen the manuscript and in this reviewer's view warrant publication in Nature Communications

<https://www.nature.com/articles/s41893-023-01201-w>

<https://pubs-rsc-org.ezp.sub.su.se/en/content/articlehtml/2018/gc/c7gc02989f>

<https://www.sciencedirect.com/science/article/pii/S1359836823001956>

We thank the reviewer for this suggestion. The mentioned works indeed offer alternative approaches towards BPA substitutes. The main advantage of these methodologies compared to the current work is avoidance of transition metal catalysis. However, these methods rely on the alkylation of the aromatic ring, which causes regioselectivity issues. E.g. in work by Sels, while the regioselectivity was significantly optimized, the mixture still consisted of the products of *ortho*, *meta*- and *para*-substitution. In case of the work by Samec, where the dimerization of ethyl guaiacol was realized via its reaction with formaldehyde, the selectivity issues might arise when the reaction is performed on more complex lignin oil, containing other guaiacol derivatives. In addition, the method developed in the presented work does not require any stoichiometric additives (e.g. formaldehyde of guaiacol).

We have now included a comparative analysis of these publications and the presented herein study. We have also included the above mentioned works as ref. 52-53. We added the following paragraph to the main text of the manuscript:

“It is worth noting that the production of bio-based BPA analogues was a subject of previous studies. Some approaches offer attractive transition metal-free pathways via e.g. zeolite-catalyzed alkylation of guaiacol with IE or ethyl guaiacol dimerization using formaldehyde. However, since those methods rely on the reactivity of an aromatic ring, the selectivity of the processes is more difficult to control, which complicates a direct application of the processes to lignin oils. In contrast, the presented herein methodology is selective towards propenyls and as such allows for the selective formation of the target bisphenols in the presence of other phenolic compounds.”

Reviewer #2 (Remarks to the Author):

The manuscript describes Lignin an alternative approach is presented, whereby the target monomers from lignin are selectively and directly converted to the final products under mild conditions and easy separation. The method the authors present is based on the hydrogen-free catalytic fractionation of wood to convert lignin into iso-eugenol (IE) and propenyl syringol (PS) enriched oil followed by an olefin metathesis step to yield bisphenols and butene-2, thus, valorising all bio-based carbons. The authors have identified experimentally that the presence of allylic alcohols as major impurities can contribute to catalyst poisoning, even when present in minor amounts. Therefore, they have developed We developed a protocol to solve this problem and improve the catalytic efficiency. The authors have also demonstrated the transformation of bisphenols to polyesters with a high glass transition temperature and good thermal stability.

The following points for the authors to consider:

1. Explain and discuss scientifically the choice of catalysts and the strategy you have chosen, as you indicate in Figure 2 focusing on the nature of structure of the desired catalyst.

We thank the reviewer for this comment. We have included the requested discussion and introduced the following modification to the main text of the manuscript:

“Several existing metathesis catalysts Hoveyda Grubbs II generation (**HGII**), nitroGrela (**nG**) and UltraNitroCat (**UNC**) were evaluated for dimerization **IE**”

was changed to:

“To enable olefin metathesis on unpurified lignin oil we decided to test ruthenium-based olefin metathesis catalysts with increased stability and rate of initiation, such as phosphine-free Hoveyda Grubbs II generation (**HGII**) and nitroGrela (**nG**). In addition, we evaluated a commercial cyclic alkyl amino carbene (CAAC)-based catalyst UltraNitroCat (**UNC**) for its known increased air and moisture tolerance. The reactions were performed with **IE** as a substrate.”

2. Figure 1, clarify which steps you present in the figure you actually study in your present work and add the acronyms in the structures you present in each figure.

We have modified Figure 1 and the caption accordingly:

“Figure 1. Strategies for upgrading of lignin-derived monomers oil. a. Previous work: Catalytic funneling of lignin-derived monomers oil, enriched in 4-propylguaiaicol and 4-propylsyringol into phenol and propylene via a two-step hydroprocessing/dealkylation. b. Previous work: Catalytic funneling of lignin-derived monomers oil enriched in 4-propanolguaiaicol and 4-propanolsyringol into 4-(3-hydroxypropyl) cyclohexan-1-ol via Raney nickel reduction and its further conversion into bio-based polyethylene terephthalate (PET) analog. PMO – porous metal oxide. c. Previous work: Functionalization of lignin-derived monomers: acryloylation of lignin-derived mixture of monomers (mainly 4-propylguaiaicol and 4-propylsyringol) and their polymerization into pressure-sensitive adhesives. d. This work: Preservation and use of inherent functionality of lignin monomers. The developed process includes: reductive catalytic fractionation of wood (RCF); hexane extraction of lignin-derived monomers oil, enriched in iso-eugenol (**IE**) and propenyl syringol (**PS**); catalytic conversion of lignin-derived monomers oil (mainly **IE** and **PS**) into bisphenols **IE-IE**, **IE-PS**, **PS-PS** (substitutes of BPA) and butene-2 via olefin metathesis; isolation of the bisphenols and their polyesterification with succinyl chloride into final materials. In addition, studies of the effect on impurities present in lignin monomers oil and their transformation into catalytically inert species were performed. BPA – bisphenol A.”

3. Have the authors optimise reaction conditions such as amount of catalyst, temperature and concentration of reactants for selecting the best experimental conditions in terms of yield of desired products.

We thank the reviewer for bringing this point up. In the case of **IE** the optimization of the reaction conditions was mainly focused on a choice of a catalyst. A comprehensive optimization of the reaction conditions on iso-eugenol (**IE**) would not be very useful for our overall goal, which is to perform the transformation on unpurified lignin oil. This is due to a significantly increased complexity of the mixture in case of the realistic substrate. We were thus satisfied with our quick screening of the conditions,

revealing that Hoveyda Grubbs II generation (**HGII**) allowed for the formation of the product in up to 85% yield at room temperature under solvent-free conditions.

We added the following clarification to the main text of the manuscript:

“We have not performed a comprehensive optimization of the reaction conditions on **IE**, which would not be directly transferable to a realistic lignin-derived mixture.”

In case of propenyl syringol (**PS**), however, we did perform an optimization presented on Figure 6, revealing that addition of solvent (toluene) positively affects the transformation, as well as addition of **IE**; screening of catalyst loading was also performed. In addition, we carried out an optimization of the reaction on realistic substrate – lignin monomers oil (LMO), which is presented on Figure 8d.

4. The authors have written that Unlike HGII and nG, UNC did not allow for the formation of the product, have they an explanation for the observed result, and if they have investigated the steric factor's role they have written as a possible reason?

We have not performed an additional experimental investigation regarding the lower reactivity of **UNC**. We, however, found support for the proposed steric reasons in literature, which we cite as ref. 34 and 35 in the main text of the manuscript.

We add the following clarification to the main text of the manuscript:

“While we have not performed any further experimental studies regarding the lower reactivity of **UNC**, based on previous literature reports, we believe steric factors might play a role in the observed catalytic behavior.”

5. In Figure 2 it would be useful for the reader to add the yield of the product depending on the choice of the catalyst. Moreover, if there are additional byproducts during the reaction they should discuss in the manuscript.

We thank the reviewer for this comment. We have modified Figure 2 accordingly (added labels with yields of the product depending on the catalyst used). The conversion was equal to the yield and no side reactions were taking place apart from the olefin metathesis. We added the following sentence to the main text of the manuscript:

“No by-products were detected in the reaction mixture and the conversion of **IE** was equal to the yield of **IE-IE**.”

6. When they study the effect of allylic alcohols on the transformation, figure 2c, the authors have studied their derivatives. Since they indicate that the quantity affects the rate of reaction, it would be useful at least for one derivate to study the effect of concentration, e.g. of cinnamyl methyl ether by varying systematically the concentration from 0 to 30%.

We thank the reviewer for bringing this point up. Allylic species such as allylic alcohols, esters and ethers are very *likely* to be present in the reaction mixture in minor amounts (1-5 mol%). However, it is very *unlikely* to find them present in larger quantities. We thus were mainly focused on their effect on the olefin metathesis within the realistic concentration range. We evaluated their effect on the

olefin metathesis of **IE** as 5 mol% additive (Figure 2c). Experiments with higher loading of allylic species are not directly relevant for the realistic feedstock. Moreover, reaction rates of olefin metathesis of **IE** in the presence of higher loadings of methyl cinnamyl ether cannot be directly compared due to: (1) higher amounts of methyl cinnamyl ether would low total catalyst loading (since now the total content of the olefinic substrates increases); (2) at this loading self-metathesis of cinnamyl methyl ether and its cross-metathesis with **IE** are taking place to a greater extent. The kinetics of these reactions is different from the self-metathesis of **IE** and as such the reaction rates cannot be directly compared.

The experiment performed with 20 mol% addition of methyl cinnamyl ether served for mechanistic understanding. The main objective of the experiment was to determine whether the catalyst decomposition occurs and whether any of the by-products are produced from methyl cinnamyl ether. We did not observe the formation of any by-products indicating no alternative non-productive pathways involving methyl cinnamyl ether under the applied conditions. Thus, we conclude that the lower reaction rate is not caused by the poisonous effect of methyl cinnamyl ether. We include the following sentence in the main text of the manuscript:

“Thus, the lower reaction rate in this case can be attributed to lower total catalyst loading (since the total amount of olefinic species increased by 20 mol%) and different kinetics of the cross- and self-metathesis of methyl cinnamyl ether.”

7. The studies of the decomposed catalysts are very important and well-presented and discussed, this is an important part of the manuscript and the presented work.

We thank the reviewer for acknowledging our efforts.

8. In the section they discuss the “self and cross-metathesis of PS” it would be useful to add the yield of the products in the table you present in Figure 6.

We thank the reviewer for this comment. We have modified the table in Figure 6 accordingly.

Entry	PS/IE, mol/mol	Conv. PS, %	Conv. IE, %	Yield (IE-IE/PS-PS/IE-PS), %	Observed ratio (IE-PS/PS-PS)	Predicted ratio for random coupling (IE-PS/PS-PS)
1	1	25	30	6/4/17	4.5	2
2	1	63	78	22/14/34	2.4	2
3	1	85	94	24/19/46	2.4	2
4	0.66	93	93	15/15/63	4.3	3
5	1.56	95	>95	30/27/40	1.5	1.3
6	3.56	20	35	15/7/6	0.78	0.56
7	3.56	47	43	15/17/13	0.77	0.56

9. A comment if the authors have studied systematically the amount of catalyst for the “self and cross-metathesis of PS” since they have written standard conditions.

We thank the reviewer for pointing this out. By standard conditions we implied the conditions developed for IE. We used this set of conditions as a starting point for optimization of the reaction on PS. The results of the optimization (catalyst loading, addition of solvent, IE/PS ratio) are presented on Figure 6. To clarify this matter, we introduced the following modification to the main text of the manuscript:

“Under standard conditions (0.01 mol% HGII, room temp., N₂ atmosphere, neat) the product was formed in 23% yield after 1 hour, after 3 hours the yield increased to 37 %, and reached 52 % after 24 hours (Figure 6a).”

was modified to:

“Under the reaction conditions developed for IE (0.01 mol% HGII, room temp., N₂ atmosphere, neat) the product was formed in 23% yield after 1 hour, after 3 hours the yield increased to 37 %, and reached 52 % after 24 hours (Figure 6a).”

10. The authors in the conclusions can comment on the reusability and stability of the catalyst based on their experimental findings, the next steps of investigation, and scale up of the process with possible challenges they envisage in the near future and if LCA studies could be used to strengthen their new methodology.

We thank the reviewer for this valid suggestion. We have introduced the following paragraphs to the discussion part of the manuscript:

“With the current experimental protocol, the reusability of the catalyst appears cumbersome. In a future effort a heterogeneous version of the catalyst can be considered. In this case mass transfer limitations will need to be addressed, due to a high viscosity of the reaction mixture at higher conversion levels.”

“Transition from the laboratory scale to pilot and eventually industrial scale will require reconsideration of several process steps. For the separation of the lignin monomers from high molecular weight fragments distillation might be a preferential approach (vs. currently implemented hexane extraction). In addition, polyesterification protocol will need further attention, where the use of succinyl chloride and dichloromethane for interfacial polymerization is avoided. With this regard, life cycle assessment (LCA) studies are pivotal.”

“The generated bisphenols PS-PS, PS-IE, and IE-IE are highly functional and can be considered as platform molecules enabling their further diversification. Development of such facile diversification methods is of interest for future studies and would allow the building of a library of the bisphenols to enable a rapid screening of their toxicity as well as material properties of the resultant polyesters.”

11. Finally, the work is very interesting and an important topic in the area of utilising lignin-derived mixtures and the authors have developed the methodology for a direct catalytic conversion of wood into polyesters under mild reaction conditions.

We are very delighted to hear such an appreciation of our work from the reviewer.

Reviewer #3 (Remarks to the Author):

The work by Alena and coworkers proposes and validates an effective and green upgrading protocol for unpurified lignin oil. Briefly, the protocol involves the selective conversion of key components (isoeugenol (IE) and propenyl syringol (PS)) into bisphenols and butene-2 via olefin metathesis, instead of chemically funneling them into a single product. The notable advantages of this strategy include mild reaction conditions, avoiding high temperatures, and the external need for a hydrogen source. They conducted a solid investigation on: 1. elucidating the key catalyst-poisoning component (allylic species) and the corresponding mechanism; 2. proposing a protocol to tackle this problem; 3. identifying the effect of the PS/IE ratio on the conversion rate. This approach facilitates the efficient isolation of final products and their subsequent polymerization into novel polyesters, demonstrating a sustainable pathway for lignin valorization with the full utilization of bio-sourced carbon. I recommend this paper be published after addressing the following comments:

1. **One of the advantages of the proposed strategy is 'all-carbon bio-sourced' and is sustainably/economically favorable. If possible, an analysis of the carbon flow of the lignin-to-polyester process is recommended, considering the broader readership of Nature Communications.**

We thank the reviewer for this valuable suggestion. We have now performed carbon flow analysis and added it to the supplementary information as Supplementary Fig. 36. We also refer to this figure in the main text of the manuscript.

Figure 36. Carbon flow analysis. The calculations are performed using the following assumptions: 1000 tons of LMO containing 72 wt% of PS and IE (PS/IE molar ratio 2.3) with addition of 60 wt% of IE relative to the total IE+PS content in LMO; 89% yield of the bisphenols in metathesis step; 64% yield of the polyesters during the polyesterification step (1.05 equiv. of succinyl chloride). The numbers on the diagram correspond to total carbon mass in the corresponding product.

2. **In the section 'Self-metathesis of IE and the effect of impurities':**

—a. **The authors mentioned that 'nG expectedly exhibited faster initiation' without providing theoretical explanations.**

We thank the reviewer for this valid comment. NitroGrela catalyst (nG) possesses a nitro group in *para* position to isopropoxy group, which results in reduced electron density on the oxygen atom of isopropoxy group (see below). This leads to a weaker coordination of the isopropoxy group to the ruthenium and as such more rapid dissociation of this group from the metal center. Since this

dissociation is directly involved in the initiation step, **nG** catalyst exhibits a faster initiation compared to Hoveyda-Grubbs 2nd generation (**HGII**), which does not contain nitro group in its structure.

We introduced the following sentence in the main text of the manuscript:

“A faster initiation in case of **nG** is caused by the electron withdrawing effect of the nitro group, which facilitates the dissociation of isopropoxy group from the metal center, which is directly involved in the initiation step.”

—b. The utilization of freshly distilled IE is proven to be more efficient than 'as purchased' IE. Please clearly indicate the purity difference.

Iso-eugenol (**IE**) was purchased from Sigma-Aldrich (98%, mixture of *cis* and *trans* isomers). **IE** is prone to oxidation upon exposure to air. This in turn leads to the formation of oxygenated products (e.g. cinneryl alcohol), which as was shown in our work causes catalyst poisoning. In addition, an oxidative dimerization of **IE** can also take place (see the structure below), this in turn would lead to the polymerization during the olefin metathesis step. We introduced the following sentence in the main text of the manuscript:

“When a freshly distilled **IE** was used the yield improved and reached 84-88%, which can be rationalized by the presence of oxygenated species in “as purchased” **IE** (Sigma-Aldrich, 98%, mixture of *cis* and *trans* isomers) acting as catalysts’ poisonings (Figure 2a).”

product of the oxidative dimerization of **IE**

—c. The 20 mol% cinnamyl methyl ether lowered the conversion after 1 hour but reached the same level as in the case of pure IE after 11 hours. Please add explanations for the possible reasons.

Introducing such a high amount of methyl cinnamyl ether lowers total catalyst loading (since now the total content of the olefinic substrates increased by 20%). In addition, at this loading self-metathesis of cinnamyl methyl ether and its cross-metathesis with **IE** are taking place to a greater extent. The

kinetics of these reactions is different from the self-metathesis of **IE** and as such the reaction rates cannot be directly compared. The main objective of this experiment was to determine whether the catalyst decomposition occurs and whether any of the by-products are produced from methyl cinnamyl ether. We did not observe the formation of any by-products indicating no alternative non-productive pathways involving methyl cinnamyl ether. Thus, we conclude that the lower reaction rate is not caused by the poisonous effect of methyl cinnamyl ether. We include the following sentence in the main text of the manuscript:

“Thus, the lower reaction rate in this case can be attributed to lower total catalyst loading (since the total amount of olefinic species increased by 20 mol%) and different kinetics of the cross- and self-metathesis of methyl cinnamyl ether.”

—d. the reaction mechanism of HGII-mediated PS/IE conversion in SI, if possible, will be helpful to understand the catalysis process and the subsequent HGII decomposition upon allylic species.

We have now added the requested mechanism to the supplementary information (Supplementary Fig. 34). We also refer to this figure in the main text of the manuscript.

3. In the section 'Mechanistic investigation of HGII decomposition by allylic species':

—a. A minor point: positioning the labels of reaction time in Fig. 3a closer to the respective spectra for clearer association.

We have now modified the figure accordingly.

—b. 'At room temperature, allyl ethers cause the decomposition of HGII at a significantly lower rate than allylic alcohols.' Please add explanations for the possible reasons.

We thank the reviewer for bringing this important question. The exact reason for that will need further investigation. According to the mechanism previously proposed in the literature (Figure 4b in the main text of the manuscript) the crucial step in the catalyst decomposition pathway is β -hydride shift followed by the reductive elimination. This step may be more favorable in case of allylic alcohols due to the formation of aldehyde via the isomerization of the initially formed vinyl alcohol. This statement, however, would need further validation via theoretical calculations.

In addition, the presence of alcohol functionality in allylic alcohols can result in alternative decomposition pathways. E.g. an exchange of the chloride ligand of the catalyst to the alkoxy group can lead to the loss of the ruthenium alkylidene via dehydrogenation (see below). Indeed, the product of dehydrogenation of cinnamyl alcohol (cinnamyl aldehyde) was observed in the reaction mixture.

We added the following paragraph to the main text of the manuscript:

“Taking into account previously proposed mechanism for the decomposition of **GI** by allylic species (Figure 4b), an increased poisonous effect of allylic alcohols vs. allylic ethers may be due to the increased propensity of allylic alcohols for β -hydride shift and/or existence of alternative decomposition pathways. E.g. the ruthenium alkylidene can be lost via dehydrogenation of cinnamyl alcohol (Supplementary Fig. 35), which is in line with the observed formation of cinnamyl aldehyde and previous reports. However, more investigations will be needed for further conclusions.”

We added the following figure to the supplementary information:

Figure 35. A possible alternative decomposition pathway of **HGII** via dehydrogenation of cinnamyl alcohol.

—c. The authors compare RuH and HGII for the tandem sequence. What is the purpose of this comparison? If it solely serves as offering an alternative catalyst while not having a firm conclusion on which one is better, would it be better to put it in SI to have a more coherent streamline in the main discussion?

This indeed was not clearly communicated in the manuscript. We discuss these two systems (**HGII** and **RuH**) because they allow for two alternative pathways for the elimination of allylic alcohols (via isomerization and disproportionation respectively). We believe that this might be important for the application of the method to different substrates, where one of the pathways (isomerization or disproportionation) can be undesirable. In addition, it is important for further development of the system, where alternative metal-free isomerization/disproportionation protocols can be developed. We add the following paragraph to the main text of the manuscript:

“Thus, we have developed two alternative systems for the elimination of allylic alcohols from the reaction mixture: via either disproportionation or isomerization. The development of metal-free protocols for these pathways is of interest for future studies.”

4. I recommend comparing the properties of lignin-derived polyester and commercially available polyester to further strengthen the practicality of this work.

We thank the reviewer for this comment, this comparison is indeed important. We have now added the following paragraph to the main text of the manuscript:

“The thermal properties of **P-IE-IE** are comparable (or even superior) to commercial polyesters (PET, $T_g = 80\text{ }^\circ\text{C}$, $T_m = 251\text{ }^\circ\text{C}$). The thermal properties of **P-PS-IE**, which exhibited high $T_g = 149.2 \pm 0.7\text{ }^\circ\text{C}$ are matching commercial amorphous BPA-based polycarbonates ($T_g = 150\text{ }^\circ\text{C}$).”

Reviewers' Comments:

Reviewer #1:

Remarks to the Author:

The authors have taken great care in answering the points raised by this reviewer who supports publication in Nature Communication.

Reviewer #2:

Remarks to the Author:

The authors have replied to the comments of the reviewers, therefore the manuscript can be accepted.

Reviewer #3:

Remarks to the Author:

My comments have been perfectly addressed. Congrats on the great work!